# Efficient Ensembling Improves Training Data Attribution

## Abstract

Training data attribution (TDA) methods aim to quantify the influence of individual training data points on the model predictions, with broad applications in data-centric AI, such as mislabel detection, data selection, and copyright compensation. However, existing methods in this field, which can be categorized as *retraining-based* and *gradient-based*, have struggled with the trade-off between computational efficiency and attribution efficacy. Retraining-based methods can accurately attribute complex non-convex models but are computationally prohibitive, while gradient-based methods are efficient but often fail for non-convex models. Recent research has shown that augmenting gradient-based methods with ensembles of multiple independently trained models can achieve significantly better attribution efficacy. However, this approach remains impractical for very large-scale applications.

In this work, we discover that expensive, fully independent training is unnecessary for ensembling the gradient-based methods, and we propose two efficient ensemble strategies, DROPOUT ENSEMBLE and LoRA ENSEMBLE, alternative to naive independent ensemble. These strategies significantly reduce training time (up to 80%), serving time (up to 60%), and space cost (up to 80%) while maintaining similar attribution efficacy to the naive independent ensemble. Our extensive experimental results demonstrate that the proposed strategies are effective across multiple TDA methods on diverse datasets and models, including generative settings, significantly advancing the Pareto frontier of TDA methods with better computational efficiency and attribution efficacy. We conduct a theoretical analysis that provides insights into the success of our empirical findings.

## 1 Introduction

Training data plays an increasingly crucial role in modern artificial intelligence (AI) models (Kaplan et al., 2020). Consequently, data-centric AI emerges as a vital paradigm, emphasizing the collection, curation, and understanding of training data. *Training Data Attribution* (TDA) is a family of methods that assess the influence of each training sample on a model's output. Numerous TDA methods have been developed and applied to a wide range of data-centric AI applications, such as mislabel detection (Koh & Liang, 2017), data selection (Engstrom et al., 2024), and copyright compensation (Deng & Ma, 2023), thereby gaining increasing popularity.

However, accurately attributing the training data influence on very large-scale AI applications remains an open challenge. Existing TDA methods can be generally categorized into two groups: *retraining-based methods* and *gradient-based methods* (Hammoudeh & Lowd, 2024). Retraining-based methods involve the systematic retraining of the model with and without specific training samples to observe changes in the model's output (Ghorbani & Zou, 2019; Jia et al., 2019; Feldman & Zhang, 2020; Ilyas et al., 2022; Wang & Jia, 2023). Such methods often require thousands of model retraining, sometimes even growing with the size of the training data, to achieve satisfactory performance, which makes them infeasible for moderately large models. Gradient-based methods, on the other hand, estimate the influence by tracking the gradients of data samples (Koh & Liang, 2017; Yeh et al., 2018; Pruthi et al., 2020). These methods are typically computationally more efficient as they do not require the training of multiple models. But empirically they can be brittle in handling complex non-convex models (Basu et al., 2020; Bae et al., 2022) and be sensitive to the randomness associated with model initialization and training dynamics (Søgaard et al., 2021).

Recent studies have shown that gradient-based TDA methods can be significantly improved by ensembling tens of models independently trained with different random seeds, leading to state-of-the-art attribution efficacy on modern neural network models (Søgaard et al., 2021; Park et al., 2023). Aggregating these independently trained models helps to mitigate the randomness introduced by training dynamics, such as random initializations and stochastic optimization. However, despite its impressive performance, scaling such a naive independent ensemble approach further for very large models remains challenging due to the significant computational costs associated with training multiple models.

In this work, we hypothesize that training models independently is unnecessary for the purpose of TDA ensemble, and we propose two efficient ensemble strategies alternative to the naive independent ensemble approach. Our first strategy, DROPOUT ENSEMBLE, is motivated by dropout (Srivastava et al., 2014), a common deep learning module initially designed to efficiently approximate ensemble. DROPOUT ENSEMBLE reduces the computational costs by replacing the independently trained models in the naive ensemble approach with multiple dropout-masked models using the same original model. Our second strategy, LoRA ENSEMBLE, is motivated by an efficient fine-tuning technique, LoRA (Hu et al., 2021). Similar to DROPOUT ENSEMBLE, LoRA ENSEMBLE reduces the computational costs by replacing the independently trained models with LoRA fine-tuned models from the same original model, which is particularly suitable for generative Transformer models (Vaswani et al., 2017).

We evaluate the proposed DROPOUT ENSEMBLE and LoRA ENSEMBLE with extensive experiments. Our experiments span various datasets (MNIST (LeCun et al., 1998), CIFAR (Krizhevsky et al., 2009), and MAESTRO (Hawthorne et al., 2018)), model architectures (Multi-Layer Perceptrons (MLP), Residual Neural Networks (ResNet) (He et al., 2016), and Transformers (Vaswani et al., 2017)), and TDA methods (TRAK (Park et al., 2023), Influence Function (Koh & Liang, 2017), and Grad-Dot/Grad-Cos (Charpiat et al., 2019)). Compared to the naive independent ensemble approach, DROPOUT ENSEMBLE and LoRA ENSEMBLE can significantly reduce the training time, serving time, and space costs by respectively up to 80%, 60%, and 80% while maintaining similar TDA efficacy.

Additionally, we present a theoretical analysis to demonstrate the effectiveness of our proposed efficient ensemble strategies over the naive independent ensembling method. Specifically, we first introduce a general ensemble scheme: *two-step ensemble estimator*, which includes both DROPOUT ENSEMBLE and LoRA ENSEMBLE. *Two-step ensemble estimator* utilizes a number of base estimators and generates several variants on top of each of these base estimators. Based on mild assumptions, our theory shows that compared to the naive ensemble estimator, *two-step ensemble estimator* can achieve smaller squared error for estimating the optimal attribution score obtained from any given TDA method.

We summarize the contributions of this work as follows:

- We show that fully independent training is not a strict requirement for effective TDA with ensembles, which opens up a promising direction for developing more efficient and effective TDA methods.
- We provide theoretical analysis that compares the TDA method with naive independent ensemble and the two novel efficient ensemble strategies: DROPOUT ENSEMBLE and LoRA ENSEMBLE.
- Our proposed DROPOUT ENSEMBLE and LoRA ENSEMBLE achieve significant efficiency improvement across a diverse range of machine learning models, datasets, and TDA methods, advancing the state-of-the-art of TDA.

## 2 Related Work

TDA methods quantify the influence of each training sample on the model predictions by assigning a TDA score to the sample. These methods can be categorized into retraining-based ones and gradient-based ones (Hammoudeh & Lowd, 2024), and our work focuses on the latter. Please refer to Appendix A for more detailed related works.

**Ensembling for gradient-based TDA methods.** Recent studies have shown the effectiveness of ensembling for improving TDA scores computed with gradient-based methods (Søgaard et al., 2021; Park et al., 2023), which mitigates their typical issues associated with non-convexity and sensitivity to randomness. Ensembling normally applies the TDA method to many independently trained models. Either averaging the

final TDA scores (Søgaard et al., 2021) or aggregating some intermediate terms for score calculation (Park et al., 2023). Besides independently trained models, Park et al. (2023) suggest that model checkpoints at different stages of a single training can be used for ensembling as well. All these ensembling methods, though effective, also require a non-trivial amount of ensembles to perform well. Empirical studies show that their TDA performance suffers significantly when ensemble size is limited (Park et al., 2023). Consequently, the ensemble size, and thus the cost associated with each ensemble, poses a significant barrier to the effective use of ensembling in current TDA methods.

## 3 Method

Following our hypothesis that independently trained models are unnecessary for ensembling TDA methods, we propose efficient ensemble strategies alternative to the naive independent ensemble.

### 3.1 Preliminaries

We start by formalizing the TDA problem and the naive independent ensemble method for TDA.

**The TDA problem.** We have a training set $\mathcal{S} = \{x_1, \ldots, x_n\}$ where each $x_i \in \mathcal{X}_{\text{train}}$, a test set $\mathcal{T} = \{x_1, \ldots, x_m\}$ where each $x_i \in \mathcal{X}_{\text{test}}$, and a trained model output function $f_\Theta$ that is parameterized by $\Theta$. Here $\mathcal{X}_{\text{train}}$ and $\mathcal{X}_{\text{test}}$ are the space of the training and test data respectively. We consider both supervised learning and generative modeling settings. For the supervised learning setting, $\mathcal{X}_{\text{train}}$ and $\mathcal{X}_{\text{test}}$ are typically identical, and each element of them corresponds to a data point with both the feature and label. For the generative modeling setting, $\mathcal{X}_{\text{train}}$ refers to the space of training data (e.g., text sequence segments for autoregressive language models) while $\mathcal{X}_{\text{test}}$ refers to the space of model generation. For the model output function $f_\Theta$, typically we will use the model learned from the training set $\mathcal{S}$ (i.e., the model parameters will be chosen as $\Theta^* = \operatorname{argmin}_\Theta \sum_{x_i \in \mathcal{S}} \mathcal{L}(x_i; f_\Theta)$ for some loss function $\mathcal{L}$). However, this is not always the case in practice (Pruthi et al., 2020). For a training set $\mathcal{S}$ and any test sample $x \in \mathcal{T}$, a TDA method $\tau$ derives the TDA scores $\tau(x, \mathcal{S}; f_\Theta) \in \mathbb{R}^n$ to quantify the influence of each training data point in $\mathcal{S}$ on the model learned from $\mathcal{S}$. More specifically, the $i$-th element, $\tau(x, \mathcal{S}; f_\Theta)_i \in \mathbb{R}$, is a real-valued score indicating the importance of $x_i$ on the model output on the test sample $x$. In the supervised classification setting, the "model output on $x$" usually refers to the loss, (log-)likelihood, or logit for the model predicting the correct class of $x$ (Koh & Liang, 2017; Park et al., 2023); in the generative setting, it typically refers to the (log-)likelihood for the model generating $x$ (Deng & Ma, 2023).

**The naive independent ensemble method for TDA.** To mitigate the randomness led by the training dynamics of non-convex deep learning models, the naive independent ensemble method first trains a set of $I$ independent models, $\{\Theta^{(i)}\}_{i=1}^I$, and then derives ensembled TDA scores $\tau_{\text{ens}}(x, \mathcal{S}; \{f_{\Theta^{(i)}}\}_{i=1}^I) \in \mathbb{R}^n$ based on the set of models.

The ensembled TDA scores could be simply obtained by averaging over the TDA scores for individual models, i.e.,

$$\tau_{\text{ens}}(x, \mathcal{S}; \{f_{\Theta^{(i)}}\}_{i=1}^I) = \frac{1}{I} \sum_{i=1}^I \tau(x, \mathcal{S}; f_{\Theta^{(i)}}). \tag{1}$$

They could also come from more sophisticated aggregation over the individual models. For example, the TRAK method (Park et al., 2023) works as the following:

$$\tau_{\text{TRAK}}(x, \mathcal{S}; \{f_{\Theta^{(i)}}\}_{i=1}^I) = \left( \frac{1}{I} \sum_{i=1}^I \mathbf{Q}_{f_{\Theta^{(i)}}} \right) \cdot \left( \frac{1}{I} \sum_{i=1}^I \phi_{f_{\Theta^{(i)}}} \left( \Phi_{f_{\Theta^{(i)}}}^\top \Phi_{f_{\Theta^{(i)}}} \right)^{-1} \Phi_{f_{\Theta^{(i)}}}^\top \right), \tag{2}$$

where $\Theta^{(i)}$ are parameters of models independently trained on the training data; $\mathbf{Q}_{f_{\Theta^{(i)}}}$ is a diagonal matrix with each diagonal element corresponding to the "one minus correct-class probability" of a training data point under model $\Theta^{(i)}$; $\phi_{f_{\Theta^{(i)}}}$ is the vector gradient of $f_{\Theta^{(i)}}(x)$ with respect to $\Theta^{(i)}$; and $\Phi_{f_{\Theta^{(i)}}}$ is the matrix of the vector gradients of $f_{\Theta^{(i)}}(x_j)$ stacked over the training samples $x_j \in S$.[1]

---

[1] Some details of the exact TRAK implementation are omitted here.

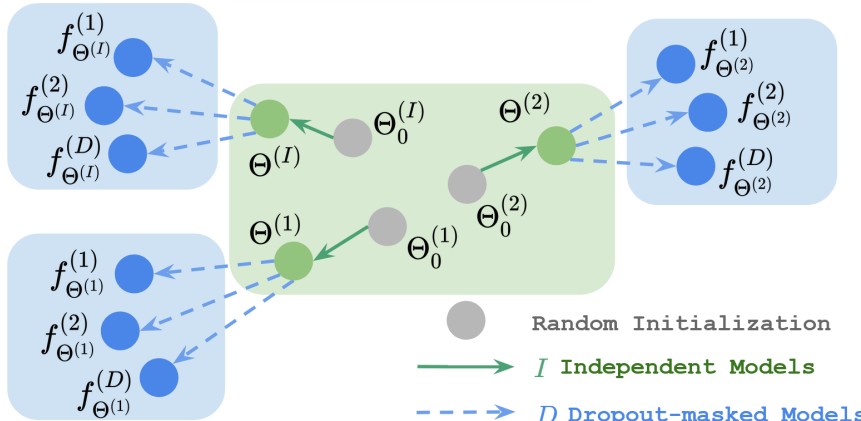

Figure 1: DROPOUT ENSEMBLE consists of two steps: (1) train $I$ models ($\{\Theta^{(i)}\}_{i=1}^I$) independently; (2) get $D$ dropout-masked models ($\{f_{\Theta^{(i)}}^{(d)}\}_{d=1}^D$) for each $i = 1, \ldots, I$.

## 3.2 Computational costs of TDA methods

While recent research has shown that ensembling gradient-based TDA methods can achieve decent attribution efficacy with tens of independently trained models (as opposed to hundreds or even thousands of model retraining in retraining-based TDA methods) (Park et al., 2023), it is still impractical for very large-scale or edge-device applications due to the increased time and space costs. To better quantify the time and space costs associated with TDA methods, we categorize the costs into three parts: training time cost, serving time cost, and storage cost. We provide a detailed explanation for these costs as follows:

- **Training time cost**: This refers to the computational time incurred by processing and training models to be used by TDA ensembles. This is the major computational challenge for ensembling TDA methods.
- **Serving time cost**: This refers to the remaining computational time to deploy TDA ensembles *after model training*. Typically, ensembling gradient-based TDA methods derives different TDA scores using multiple trained models and then aggregates them. The serving time cost will include all computational costs incurred during this process.
- **Space cost**: This refers to the parameters of the trained models that are stored by TDA ensembles. These model parameters are required to run forward and backward passes on top of the models for TDA score calculation. The space costs are generally proportional to the model size.

## 3.3 Dropout Ensemble

**General methodology.** Dropout, a common module used in many modern deep learning models, is initially designed as an efficient approximation of ensemble (Srivastava et al., 2014). Motivated by this idea, we propose a simple, efficient ensemble strategy, DROPOUT ENSEMBLE, for TDA methods. Instead of performing the TDA ensemble over a large number of independently trained models, the proposed method utilizes multiple dropout masks on the same model to perform the TDA ensemble.

In practice, DROPOUT ENSEMBLE consists of two steps as illustrated in Figure 1. In the first step, we train $I$ independent models, $\{\Theta^{(i)}\}_{i=1}^I$, as shown by the green arrows in Figure 1. Next, as shown by the blue arrows in Figure 1, for each model $\Theta^{(i)}, i \in \{1, \ldots I\}$, we obtain $D$ variants of the model with different dropout masks, which are denoted as $\{f_{\Theta^{(i)}}^{(d)}\}_{d=1}^D$. For any TDA method, DROPOUT ENSEMBLE calculates the ensembled TDA scores through $\tau_{\text{ens}}(x, \mathcal{S}; \{f_{\Theta^{(i)}}^{(d)}\}_{1 \leq i \leq I, 1 \leq d \leq D})$.

In comparison to the naive independent ensemble method, DROPOUT ENSEMBLE can **significantly reduce the training time and space costs**. As we will show in Section 4.2, this method allows us to replace the expensive independently trained models with dropout-masked models that incur no additional training cost or model parameters. There is some serving time cost overhead caused by DROPOUT ENSEMBLE . As we will

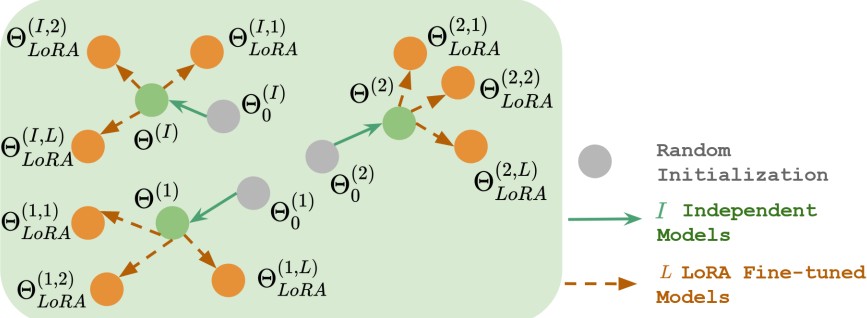

Figure 2: LoRA Ensemble consists of two steps: (1) train $I$ models ($\{\Theta^{(i)}\}_{i=1}^I$) independently; (2) get $L$ LoRA fine-tuned models ($\{\Theta_{\text{LoRA}}^{(i,l)}\}_{l=1}^L$) for each $i = 1, \ldots, I$.

show in Appendix L, the overhead is small and Dropout Ensemble could achieve better efficacy-efficiency trade-off.

**TDA-method-specific optimization.** It is possible to further optimize the proposed ensemble strategy when applying it to a particular TDA method. For example, we develop a variant of Dropout Ensemble tailored for the TRAK method to reduce the serving time cost of Dropout Ensemble. Recall that in Eq. (2), the quantities $Q$'s only involve the forward pass of the models on the training data, while $\phi$'s and $\Phi$'s require the backward pass on the training data. We find that, perhaps a bit surprisingly, if we only use the dropout-masked models to calculate $Q$'s while using the original models to calculate $\phi$'s and $\Phi$'s, we can get similar attribution efficacy. Concretely, directly applying Dropout Ensemble on TRAK leads to

$$\tau_{\text{ens}}\left(x, \mathcal{S}; \{f_{\Theta^{(i)}}^{(d)}\}_{\substack{1 \leq i \leq I \\ 1 \leq d \leq D}}\right) = \left(\frac{1}{I \cdot D}\sum_{i=1}^I\sum_{d=1}^D \mathbf{Q}_{f_{\Theta^{(i)}}^{(d)}}\right) \cdot \left(\frac{1}{I \cdot D}\sum_{i=1}^I\sum_{d=1}^D \phi_{f_{\Theta^{(i)}}^{(d)}}\left(\Phi_{f_{\Theta^{(i)}}^{(d)}}^\top \Phi_{f_{\Theta^{(i)}}^{(d)}}\right)^{-1}\Phi_{f_{\Theta^{(i)}}^{(d)}}^\top\right),$$

while in the optimized version, we have

$$\tau_{\text{ens}}\left(x, \mathcal{S}; \{f_{\Theta^{(i)}}^{(d)}\}_{\substack{1 \leq i \leq I \\ 1 \leq d \leq D}}, \{f_{\Theta^{(i)}}\}_{1 \leq i \leq I}\right) = \left(\frac{1}{I \cdot D}\sum_{i=1}^I\sum_{d=1}^D \mathbf{Q}_{f_{\Theta^{(i)}}^{(d)}}\right) \cdot \left(\frac{1}{I}\sum_{i=1}^I \phi_{f_{\Theta^{(i)}}}\left(\Phi_{f_{\Theta^{(i)}}}^\top \Phi_{f_{\Theta^{(i)}}}\right)^{-1}\Phi_{f_{\Theta^{(i)}}}^\top\right).$$

In this case, we only need to evaluate the forward pass on the dropout-masked models and reduce the serving time cost by avoiding the backward pass. We call this variant of our method as Dropout Ensemble (forward-only).

### 3.4 LoRA Ensemble

For large-scale generative models, especially Transformer models (Vaswani et al., 2017), LoRA adapters have shown great success for efficiently fine-tuning the models (Hu et al., 2021). Our second efficient ensemble strategy, LoRA Ensemble, is motivated by the LoRA techniques. In particular, we propose to replace the independently trained models in the naive ensemble method with LoRA fine-tuned models.

Similar to Dropout Ensemble, LoRA Ensemble also consists of two steps as illustrated in Figure 2. The first step trains $I$ independent models, $\{\Theta^{(i)}\}_{i=1}^I$ (green arrows in Figure 2), while the second step further trains $L$ LoRA fine-tuned models, $\{\Theta_{\text{LoRA}}^{(i,l)}\}_{l=1}^L$, for each $\Theta^{(i)}$, $i \in \{1, \ldots, I\}$ (brown arrows in Figure 2).

In comparison to Dropout Ensemble, LoRA Ensemble will introduce a small amount of additional training time cost and model parameters for LoRA fine-tuning. However, since we only use the parameters in LoRA adapters to compute the TDA scores, LoRA Ensemble can reduce the serving time cost compared to either dropout-masked models in Dropout Ensemble or independent models in the naive method. This leads to a unique advantage for LoRA Ensemble if the serving time cost is critical among the computational costs.

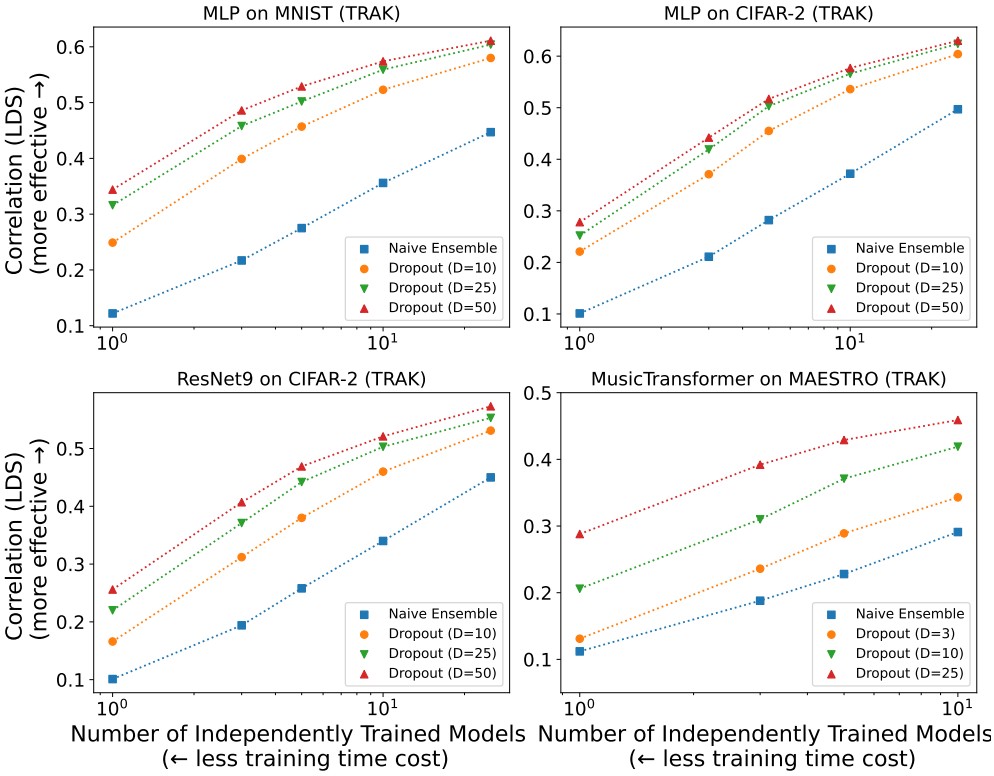

Figure 3: The LDS of naive independent ensemble and Dropout Ensemble with different numbers of dropout-masked passes ($D$) and independently trained models ($I$). We apply the ensemble methods to the TDA method, TRAK. There are four experiment settings: MLP classifiers trained on MNIST and CIFAR-2 (top row); ResNet9 trained on CIFAR-2 (bottom-left); and Music Transformer trained on MAESTRO (bottom-right). The $x$-axis indicates the training time cost measured by the number of independently trained models ($I$). The $y$-axis indicates the attribution efficacy measured by LDS.

## 4  Experiments

In this section, we empirically evaluate the efficiency and efficacy of the proposed Dropout Ensemble and LoRA Ensemble.

### 4.1  Experimental setup

We conduct extensive experiments on a wide range of settings.

**TDA methods.**   We apply the proposed methods to various gradient-based TDA methods, including influence function (IF) (Koh & Liang, 2017), Grad-Dot/Grad-Cos (Charpiat et al., 2019), and TRAK (Park et al., 2023). The detailed descriptions and implementations of these algorithms are provided in Appendix B.

**Datasets and models.**   We consider models with different architectures trained on diverse datasets: (1) a three-layer MLP classifier trained on the MNIST-10 dataset (LeCun et al., 1998), (2) a three-layer MLP classifer and a ResNet-9 classifier (He et al., 2016) trained on the CIFAR-2 dataset (a two-class subset of the CIFAR-10 dataset (Krizhevsky et al., 2009)), and (3) a Music Transformer (Huang et al., 2018) trained on the MAESTRO dataset (Hawthorne et al., 2018). Notably, the former two are supervised classification settings, while the last one is a generative modeling setting. We sample 5000 training samples and 500 test samples from MNIST-10 and CIFAR-2 datasets. For MAESTRO dataset, we sample 5000 training samples and 178 generated samples. More detailed setups for each dataset and model are listed in Appendix C. MNIST-10

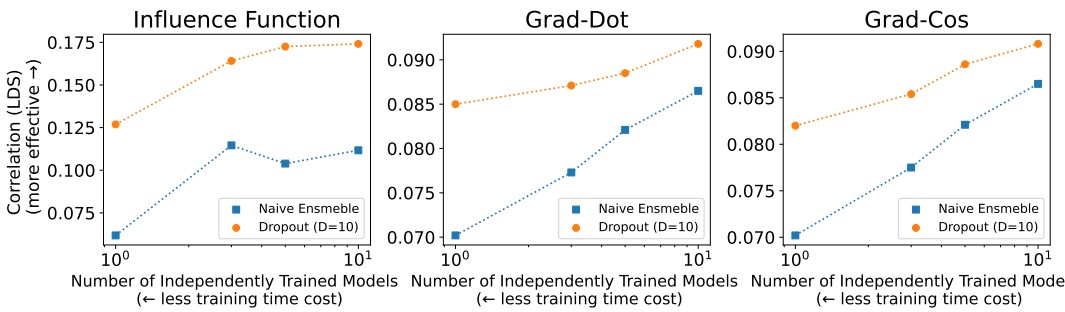

Figure 4: The LDS of naive independent ensemble and DROPOUT ENSEMBLE on more TDA methods, IF, Grad-Dot, and Grad-Cos. The experiments are performed on MLP classifiers trained on MNIST. The plot setup is similar as Figure 3.

dataset holds CC BY-SA 3.0 license. CIFAR-10 dataset holds CC-BY 4.0 license. MAESTRO dataset holds CC BY-NC-SA 4.0 license.

**Evaluation metric for TDA efficacy.** We utilize the linear datamodeling score (LDS) (Park et al., 2023) to evaluate the efficacy of the TDA methods augmented by different ensembling methods. Intuitively, LDS measures the rank correlation between the TDA scores among training samples and the change of model outputs by removing a subset of training samples and retraining the model from scratch. A higher LDS value corresponds to a better alignment between the TDA scores and the influence of the training samples on the model outputs, thus better TDA quality. We refer the reader to Appendix E for the exact definition of LDS.

**Evaluation metrics for TDA efficiency.** As introduced in Section 3.2, we measure the TDA efficiency in terms of the training time cost, serving time cost, and space cost. For both training and serving time costs, we measure the wall-clock time on a single A40 GPU. For the space cost, although ideally one would measure it by memory usage, this can be challenging because memory usage is sensitive to the specific implementation and parallelization. Therefore, we measure the space cost by the total parameter count instead. For DROPOUT ENSEMBLE, we will also use the number of independently trained models ($I$) as a measure of the training time cost and space cost, as both of the two costs are proportional to $I$ in this case. More details about the measurements are provided in Appendix F.

## 4.2 Dropout Ensemble

**Improvement over the training time cost.** To illustrate the improvement over training time cost comparing DROPOUT ENSEMBLE to the naive ensemble, we first report the results on the TRAK method across four experiment settings. As can be seen in Figure 3, across all four experiment settings, increasing the number of dropout-masked passes ($D$) results in significant LDS improvement for a fixed number of independently trained models ($I$). Recall that DROPOUT ENSEMBLE does not incur any additional training time cost for a fixed $I$. This implies that DROPOUT ENSEMBLE can significantly reduce the training time cost for achieving the same level of attribution efficacy as measured by LDS. For example, DROPOUT ENSEMBLE with $I = 5$ can reach higher LDS than naive independent ensemble with $I = 25$ for MLP on MNIST and MLP/ResNet9 on CIFAR-2, which achieves a 80% reduction on training time cost. For Music Transformer on MAESTRO, DROPOUT ENSEMBLE with $I = 1$ can reach higher LDS than naive independent ensemble with $I = 10$, which achieves a 90% reduction.

We find that DROPOUT ENSEMBLE works similarly well for other TDA methods. In Figure 4, we report the results on IF, Grad-Dot, and Grad-Cos. We only experiment on the setting of MLP classifiers trained on MNIST, as these TDA methods do not have meaningfully good efficacy on more complex settings. Similar to the TRAK experiments, we observe that DROPOUT ENSEMBLE with $I = 1$ could match the LDS of naive independent ensemble with $I = 10$ for IF and Grad-Dot, and match that with $I = 5$ for Grad-Cos. Therefore, we expect that the proposed DROPOUT ENSEMBLE can be generalized to various gradient-based TDA methods.

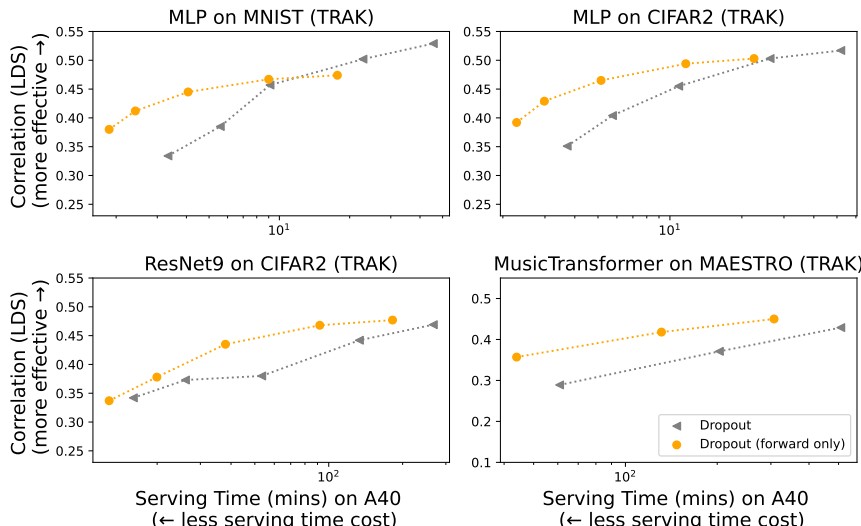

Figure 5: The LDS of DROPOUT ENSEMBLE and its variant DROPOUT ENSEMBLE (forward-only) when applied to TRAK on different dataset and model settings. The $x$-axis indicates the serving time cost measured by the running time on a single A40 GPU. The $y$-axis indicates the attribution efficacy measured by LDS. The points in the plot correspond to different numbers of dropout masked models ($D$). The number of independently trained models ($I$) is fixed to 5.

**Improvement over the space cost.** The improvement of DROPOUT ENSEMBLE over the space cost is self-evident, which can also be measured by $I$, the number of independently trained models. According to Figure 3, for example, applying DROPOUT ENSEMBLE on TRAK (in comparison to naive independent ensemble) can lead to a 80% or 90% reduction in space cost for different model and dataset settings.

**Dropout Ensemble (forward-only) improves the serving time cost.** Figure 5 shows the results of DROPOUT ENSEMBLE (forward-only), a variant of DROPOUT ENSEMBLE specifically tailored for TRAK. In comparison to the vanilla DROPOUT ENSEMBLE, this optimized variant can achieve comparable TDA accuracy with around 50% less serving time.

We also report additional ablation studies and results of DROPOUT ENSEMBLE in Appendix G as well as results on larger dataset and models in Appendix K.

### 4.3 LoRA Ensemble

To demonstrate the efficiency of LoRA ENSEMBLE, we only consider TRAK Park et al. (2023) as the main TDA method in this section since it is the most effective (in terms of LDS) among the four gradient-based TDA methods discussed in Section 4.2. We experiment LoRA ENSEMBLE under the setting of Music Transformer on MAESTRO because this setting aligns better with large-scale real-world generative settings where LoRA adapters are more prevalent.

As shown in Figure 6, LoRA ENSEMBLE outperforms naive independent ensemble since it can reach similar LDS as the latter with significantly less computational costs in terms of all types of measurements. For example, LoRA ENSEMBLE with ($I = 1, L = 3$) achieves similar LDS as naive ensemble with $I = 5$, while having reduction in training time, serving time, and total parameter count respectively for 78.4%, 60.5%, and 79.6%.

We also report additional results on larger dataset and models in Appendix K.

## 5 Theoretical Analysis of Naive and Two-step Ensemble

In this section, we present a theoretical analysis to compare TDA method with the naive ensemble (regular independent ensemble) and our two-step ensemble (a general case containing DROPOUT ENSEMBLE and

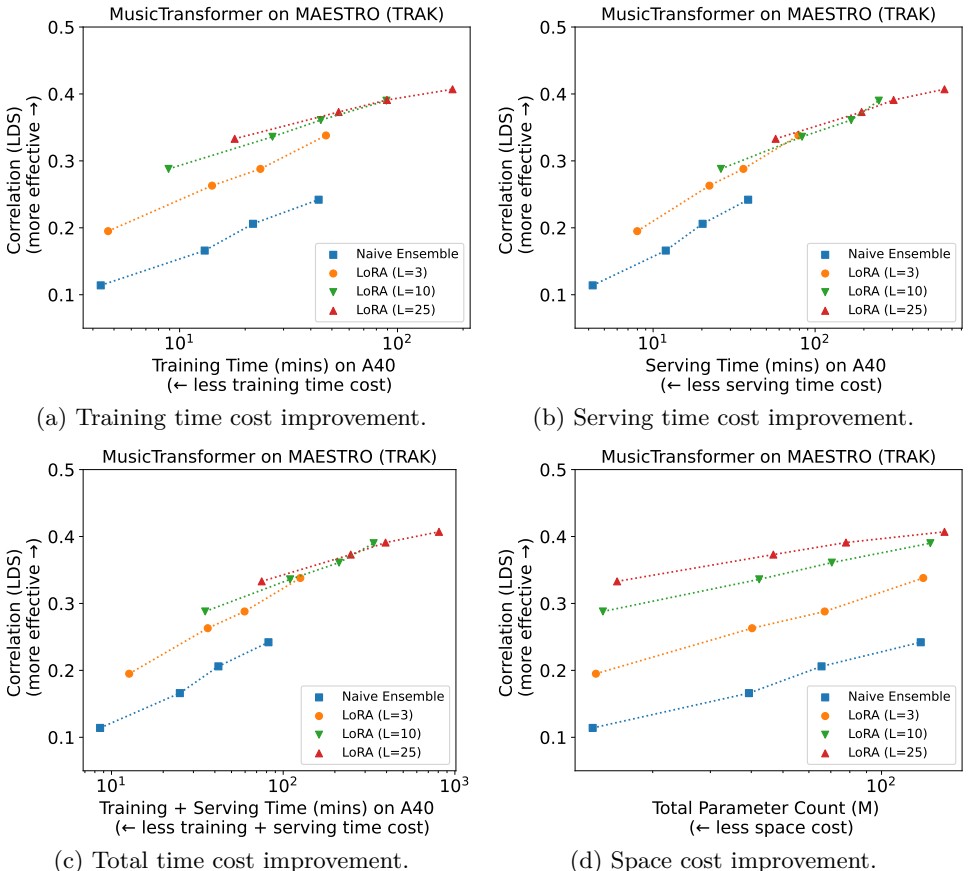

(a) Training time cost improvement.

(b) Serving time cost improvement.

(c) Total time cost improvement.

(d) Space cost improvement.

Figure 6: The LDS of naive ensemble and LoRA ENSEMBLE with respect to different cost measurements. Here, we apply the ensemble methods to TRAK on Music Transformer trained on the MAESTRO dataset. The x-axis of Figure (6a, 6b, 6c) indicates training/serving time costs (running time on a single A40), while that of Figure 6d specifies the space cost (total parameter count). The y-axis of all figures is the attribution efficacy measured by LDS. LoRA ENSEMBLE has significantly fewer costs in all aspects than naive ensemble for achieving similar LDS.

LoRA ENSEMBLE). With notation defined in Section 3.1, for any test sample $x \in \mathcal{T}$ and $\mathcal{S}$ as the training set, we assume a TDA method $\tau$ finds an "optimal" attribution score $\tau(\Theta^*)$ based on the optimal parameter $\Theta^*$ that some ensemble estimators try to approximate (omit $f_\Theta$ here for notational convenience). Then, we define the two types of ensemble estimators for $\tau(\Theta^*)$ and through Lemma 5.1 below to show that DROPOUT ENSEMBLE and LoRA ENSEMBLE outperform the regular ensemble in terms of the approximation error.

We start with the definition of the two types of ensemble estimators. For a TDA method $\tau$, the *regular ensemble estimator* $\tau_{ens}$ is defined as the average of $\tau(\Theta^{(i)})$, where $\Theta^{(i)}$ for $i = 1, \ldots, I$ are $I$ identically distributed (i.d.) individual estimators. The *two-step ensemble estimator* $\tau_{2\text{-step}}$ is defined as the average of $\tau(\Theta^{(k,d)})$, where each of $\Theta^{(k)}$ for $k = 1, \cdots, K$ is an individual estimator (just as $\Theta^{(i)}$ in *regular ensemble estimator*) and by using $\Theta^{(k)}$ as a base estimator, $D$ variants of it, $\Theta^{(k,d)}$, are generated for the second-step ensemble. This is a general definition and includes DROPOUT ENSEMBLE and LoRA ENSEMBLE as special cases. Our goal is to compare the squared error of $\tau_{ens}$ and $\tau_{2\text{-step}}$ with respect to $\tau(\Theta^*)$, which we formalize as the following lemma (proof in Appendix J.1).

**Lemma 5.1.** *Define $\tau_{ens}$ and $\tau_{2\text{-step}}$ as the regular and two-step ensemble estimators for $\tau(\Theta^*)$:*

$$\tau_{ens} = \frac{1}{I} \sum_{i=1}^{I} \tau(\Theta^{(i)}) \quad and \quad \tau_{2\text{-step}} = \frac{1}{KD} \sum_{k=1}^{K} \sum_{d=1}^{D} \tau(\Theta^{(k,d)}).$$

*Assume each individual estimate $\tau(\Theta^{(i)})$ and $\tau(\Theta^{(k,d)})$ have the same distribution. Then, the difference in squared error $\Delta = \|\tau_{ens} - \tau(\Theta^*)\|^2 - \|\tau_{2\text{-}step} - \tau(\Theta^*)\|^2$ is given by:*

$$\Delta = \frac{1}{I}\left(\Sigma_{1,1} + (I-1)\Sigma_{i,j}\right) - \frac{1}{KD}\bigg(\Sigma_{(k,m),(k,m)} +$$

$$(D-1)\Sigma_{(k,m),(k,n)} + D(K-1)\Sigma_{(k,m),(l,n)}\bigg),$$

*where:*

- $\Sigma_{i,j} = Cov(\tau(\Theta^{(i)}), \tau(\Theta^{(j)}))$ *is the covariance between individual estimators in the regular ensemble. Given the i.d. assumption, $\Sigma_{i,j}$ are the same for all $i \neq j$, and $\Sigma_{i,j} = \Sigma_{1,1}$ for $i = j$.*
- $\Sigma_{(k,m),(l,n)} = Cov(\tau(\Theta^{(k,m)}), \tau(\Theta^{(l,n)}))$ *is the covariance between variants of base individual estimators in the two-step ensemble. We further define $\Sigma_{(k,m),(k,n)}$ as the within-group covariance and $\Sigma_{(k,m),(l,n)}$ with $k \neq l$ as the between-group covariance.*

Now, we analyze $\Delta$ and show that $\Delta$ will stay positive under reasonable assumptions, which implies that $\tau_{2\text{-step}}$ outperforms $\tau_{ens}$. We especially derive $\Delta$ for an interesting cases: $K = I$ (see assumpions and derivation in Appendix J.3), and we discuss another case: $KD = I$ in Appendix J.2.

**Case 1:** $K = I$ means the number of individual estimators $K$ in the two-step ensemble is equal to the number of individual estimators $I$ in the regular ensemble, but the two-step ensemble has $D$ variants for each base estimator, resulting in a total of $K \cdot D = I \cdot D$ estimates. In this case, we drive

$$\Delta = \frac{D-1}{ID}\left(\Sigma_{1,1} - \Sigma_{(k,m),(k,n)}\right). \tag{3}$$

As long as the within-group covariance $\Sigma_{(k,m),(k,n)}$ is smaller than variance of each individual estimator $\Sigma_{1,1}$, $\tau_{2\text{-step}}$ will outperform the regular ensemble $\tau_{ens}$. This is likely to be the case for any pair of different variants of the same base estimator through dropout or LoRA fine-tuning. For a fixed small $I$, when $D$ is small, $\frac{D-1}{D}$ has bigger impact on $\Delta$ and increase $D$ can quickly increase $\Delta$ means $\tau_{2\text{-step}}$ outperforms $\tau_{ens}$ more. In contrast, when $D$ is large, $\frac{D-1}{D}$ is close to 1 and increase $D$ will not change $\Delta$ much, meaning the performance gain of $\tau_{2\text{-step}}$ over $\tau_{ens}$ will saturate, which matches our empirical results in Figure 3 and Figure 6. On the other hand, for a large $I$, changing $D$ will not change $\Delta$ much as $\Delta$ is already small, and the performance gain of $\tau_{2\text{-step}}$ over $\tau_{ens}$ will be less significant, which also aligns with our empirical observation in Figure 3 and Figure 6. In summary, when $K = I$, $\tau_{2\text{-step}}$ clearly outperforms $\tau_{ens}$ when the within-group covariance is small benefiting from averaging over more total estimators ($ID$ versus $I$), and the performance gain saturates as the within-group covariance increases.

## 6 Conclusion

We present DROPOUT ENSEMBLE and LORA ENSEMBLE as efficient alternatives to the naive independent ensemble approach for improving gradient-based TDA methods. The proposed strategies significantly reduce training time (up to 80%), serving time (up to 60%), and space cost (up to 80%), while maintaining similar attribution efficacy in comparison to the naive ensemble. Empirical results from our extensive experiments show that the proposed efficient ensembles can remarkably advance the Pareto frontier of TDA methods with better computational efficiency and TDA efficacy. Notably, we demonstrate the proposed methods work well on a generative modeling setting. Our theoretical analysis shows our ensemble method has a smaller squared error for estimating the optimal scores compared to naive ensembles. In the future, we will utilize our methods in more real-world generative settings that were blocked by high computational costs or low effectiveness of TDA methods.

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

## A More related work

**Retraining-based TDA methods.**  Retraining-based methods compute TDA scores by systematically retraining the model with and without specific training samples to quantify their influence on the model's output. These methods are computationally expensive due to the requirement of the large amount of model retrainings. For example, Leave-One-Out (LOO) influence Tukey (1958) measures the prediction difference between the model trained on the full training dataset and models trained on subsets with only one specific sample dropped. Data Shapley (Ghorbani & Zou, 2019; Jia et al., 2019), Beta-Shapley (Kwon & Zou, 2022), and Data Banzhaf (Wang & Jia, 2023) extends the LOO idea to consider data interactions for more equitable TDA scores, but they require retraining models on all possible subsets of the training dataset. Similarly, DataModels (Ilyas et al., 2022) tries to learn the model predictions when the model is trained on each subset of the training dataset, which requires a nontrivial amount of model retraining. While, in practice, sampling or approximation will be used to reduce the number of model retrainings, these methods typically still require thousands of or more retrainings to achieve satisfactory attribution efficacy. In summary, the high demand for retraining makes these TDA methods computationally inefficient and limits their practical applicability to even moderately large models.

**Gradient-based TDA methods.**  Gradient-based methods are another group of TDA methods that usually provide closed-form TDA scores using gradients. Since the seminal work of influence function by Koh & Liang (2017), gradient-based methods have become increasingly popular due to their scalability. The influence function (Koh & Liang, 2017) and subsequent studies (Guo et al., 2021; Barshan et al., 2020; Schioppa et al., 2022; Kwon et al., 2023) obtain TDA scores by approximating the effect of upweighting a training sample on the loss function. Moreover, Representer Point Selection (Yeh et al., 2018) decomposes the pre-activation of a neural network as a linear combination of training samples. TracIn (Pruthi et al., 2020) traces the loss changes on the test points during the training process. TRAK (Park et al., 2023) uses the neural tangent kernel with random projection to assess influence. These gradient-based methods significantly reduced the computational cost compared to retraining-based methods. The downside is that they typically rely on the convexity assumption and Taylor approximation to calculate TDA scores. These requirements lead to performance degradation on non-convex neural networks and sensitivity to the randomness inherent in model initialization and training. To improve the efficiency of these TDA methods, grouping training data has been explored (Ley et al., 2024).

**Dropout and LoRA in ensembling.**  There have been some studies using dropout (Gal & Ghahramani, 2016) and LoRA (Wang et al., 2023) in ensembling, though not connected with TDA. To the best of our knowledge, this is the first study to propose an alternative to fully independent training to achieve a better balance between effectiveness and efficiency in TDA ensembling. Additionally, it uncovers intriguing links between dropout, LoRA fine-tuning, and ensembling within the TDA framework. Moving beyond conventional ensembling is both an interesting and practically valuable discovery.

## B TDA methods

In this section, we provide details on the TDA methods we perform efficient ensembling on. Additionally, we also introduce the ensembling aggregation methods, i.e., the way to aggregate $\tau_{\text{ens}}(x, \mathcal{S}; \{f_{\Theta^{(i)}}\}_{i=1}^{I})$, for each method. The notation will follow Section 3.1.

**Tracing with the Randomly-projected After Kernel (TRAK).**  This is a state-of-the-art gradient-based TDA method provided by Park et al. (2023). It natively introduces ensembling in its definition.

$$\tau_{\text{TRAK}}(x, \mathcal{S}; \{f_{\Theta^{(i)}}\}_{i=1}^{I}) = \left( \frac{1}{I} \sum_{i=1}^{I} \mathbf{Q}_{f_{\Theta^{(i)}}} \right) \left( \frac{1}{I} \sum_{i=1}^{I} \phi_{f_{\Theta^{(i)}}} \left( \Phi_{f_{\Theta^{(i)}}}^{\top} \Phi_{f_{\Theta^{(i)}}} \right)^{-1} \Phi_{f_{\Theta^{(i)}}}^{\top} \right),$$

Detailed notation description is described in Section 3.1. We use 2048 as the random projection dimension for TRAK. For each independently trained model, we train the model on half sampled training set following Park et al. (2023).

**Influence functions based on the conjugate gradients (IF).** First proposed by Koh & Liang (2017), the definition of IF is

$$\tau_{\text{IF}}(x, \mathcal{S}; \{f_{\Theta^{(i)}}\}_{i=1}^I) = \left[\frac{1}{I}\sum_{i=1}^I g_{f_{\Theta^{(i)}}}(x_j)^\top H_{f_{\Theta^{(i)}}}^{-1} g_{f_{\Theta^{(i)}}}(x) : x_j \in \mathcal{S}\right],$$

where $g_{f_{\Theta^{(i)}}}(x)$ is the vector gradient of model output $f(x; \Theta^{(i)})$ with respect to the parameters $\Theta^{(i)}$, $H_{f_{\Theta^{(i)}}}^{-1}$ is the inverse hessian matrix with respect to the training set, and $g_{f_{\Theta^{(i)}}}(x_j)^\top$ is the vector gradient to the training sample. The product of the first two terms are an inverse-hessian-vector-product problem, we implement conjugate gradients approach to solve it.

**Grad-Dot.** Grad-Dot is proposed by Charpiat et al. (2019). We simply calculate Grad-Dot multiple times on trained parameters $\Theta^{(i)}, i \in \{1, \ldots, K\}$ and take the average value.

$$\tau_{\text{Grad-Dot}}(x, \mathcal{S}; \{f_{\Theta^{(i)}}\}_{i=1}^I) = \left[\frac{1}{I}\sum_{i=1}^I g_{f_{\Theta^{(i)}}}(x)^\top g_{f_{\Theta^{(i)}}}(x_j) : x_j \in \mathcal{S}\right],$$

where $g_{f_{\Theta^{(i)}}}(x)^\top$ is the vector gradient of model output $f(x; \Theta^{(i)})$ with respect to the parameters $\Theta^{(i)}$, and $g_{f_{\Theta^{(i)}}}(x_j)^\top$ is the gradient to the training sample.

**Grad-Cos.** Similar to Grad-Dot,

$$\tau_{\text{Grad-Cos}}(x, \mathcal{S}; \{f_{\Theta^{(i)}}\}_{i=1}^I) = \left[\frac{1}{I}\sum_{i=1}^I \frac{g_{f_{\Theta^{(i)}}}(x)^\top}{\|g_{f_{\Theta^{(i)}}}(x)\|} \frac{g_{f_{\Theta^{(i)}}}(x_j)}{\|g_{f_{\Theta^{(i)}}}(x_j)\|} : x_j \in \mathcal{S}\right],$$

where $g_{f_{\Theta^{(i)}}}(x)^\top$ is the vector gradient of model output $f(x; \Theta^{(i)})$ with respect to the parameters $\Theta^{(i)}$, and $g_{f_{\Theta^{(i)}}}(x_j)^\top$ is the gradient to the training sample.

## C   Detailed experiment setup

**MLP on MNIST-10.** For the MNIST-10 (LeCun et al., 1998) experiment, we sampled 5000 training samples and 500 testing samples. We used a 3-layer MLP with hidden layer sizes equal to 128 and 64 and placed dropout layers after the first two linear layers with a rate of 0.1, which resulted in a total of about 0.11M parameters. We employed an SGD optimizer with learning rate 0.01, momentum 0.9, and batch size 64 to train this MLP classifier for 100 epochs.

**MLP/ResNet-9 on CIFAR-2.** For CIFAR-2 experiment, we construct the CIFAR-2 dataset by sampling from the CIFAR-10 dataset that only include the "cat" and "dog" classes (same as the setting in Park et al. (2023)). To incorporate a diverse set of model architectures, we train an MLP and a CNN-based model on this dataset. We also only consider a subset of the CIFAR-2 dataset with a training size equal to 5000 and a testing size equal to 500. Firstly, we use a 3-layer MLP with hidden layer sizes equal to 120 and 84 and place dropout layers after the first two linear layers with a rate of 0.1, which results in a total of about 0.38M parameters. We employ an SGD optimizer with a learning rate of 0.01, momentum of 0.9, and batch size 64 to train this MLP classifier for 50 epochs. Additionally, we consider a standard ResNet-9 model (He et al., 2016) with dropout layers being placed after all convolution layers, which has roughly 4.83M trainable parameters. We train this model for 50 epochs.

**MusicTransformer on MAESTRO.** For the MAESTRO experiment, we use the MIDI and Audio Edited for Synchronous TRacks and Organization (MAESTRO) dataset (v2.0.0) Hawthorne et al. (2018) and construct a Music Transformer following the original setting in (Huang et al., 2018). Specifically, the number of layers equals to 6, the number of independent heads equal to 8, the input feature size is 512 and the dimension of the feedforward network is 1024. For data processing, we follow the basic experiment setup

used by (Deng & Ma, 2023). To be more specific, we define a vocabulary set of size equal to 388, which includes "NOTE ON" and "NOTE OFF" events for 128 different pitches, 100 "TIME SHIFT" events, and 32 "VELOCITY" events. The raw data is pre-processed as sequences of about 90K events. Due to computational constraints, we train a Music Transformer model on a subset of the official training set in the MAESTRO dataset with a size of 5000. For training, the batch size has been set to 64, and the model is trained by a classic seq2seq loss function. We employ an Adam optimizer with a learning rate equal to 1e-4, $\beta_1 = 0.9$ and $\beta_2 = 0.98$. We apply zero warm-up steps since the dataset size is comparably small, and we train the model for 20 epochs. For music event generation, we use 178 samples from the official testing dataset as prompts to generate music with a single event, which is used to evaluate the TDA methods.

## D    Efficient ensemble setup

### D.1    Dropout Ensemble

Dropout is applied to the same layers as the model training stated in Appendix C. Dropout rate is set to 0.1 for all experiment settings.

### D.2    LoRA Ensemble

This method is only applied to MusicTransformer trained on the MAESTRO dataset. According to Section 3.4, we first train $I$ independent models on the full training dataset with 10 epochs using different model initialization, and all the other remaining settings follow Appendix C. The LoRA adapters used in this experiment all have rank $r = 8$, alpha $\alpha = 8$, and trainable biases. No dropouts are applied for LoRA parameters. We augment LoRA adapters to only the $W_q$ and $W_v$ matrices in the self-attention modules within all the layers of the MusicTransformer. Then, we define fine-tuning datasets for each LoRA adapter as random training data subsets with a size equal to 2500 (i.e., half of the original training dataset size). We further fine-tune each of the aforementioned $I$ independent models using $L$ different LoRA adapters for an additional 10 epochs on these random training data subsets, which results in a total of $I \cdot L$ LoRA fine-tuned models.

## E    Linear datamodeling score (LDS)

Park et al. (2023) proposed the *linear datamodeling score* (LDS), aiming at probing the TDA method's ability to make counterfactual predictions based on the attribution score derived from the learned model output function $f_\Theta$ and the corresponding dataset to train $f_\Theta$. Because most TDA methods are assumed to be *additive*[2], the TDA scores can be used to predict the model output function learned from a subset of training data in a summation form. Formally, the *attribution-based output predictions* of the model output function $f_{\Theta_{\mathcal{S}'}}$ is defined as follows:

$$g_\tau(x, \mathcal{S}'; \mathcal{S}) \triangleq \sum_{i:x_i \in \mathcal{S}'} \tau(x, \mathcal{S}; f_\Theta)_i, \tag{4}$$

where $\mathcal{S}$ is the training set, $\mathcal{S}' \subseteq \mathcal{S}$ is a subset of $\mathcal{S}$ and $f_{\Theta_{\mathcal{S}'}}$ is the model output function with $\Theta_{\mathcal{S}'}$ learned from $\mathcal{S}'$. Intuitively, $g_\tau(x, \mathcal{S}'; \mathcal{S})$ computes the overall attribution of the subset $\mathcal{S}'$ on example $x$, which should be a powerful indicator of the model prediction on $x$ (i.e., $f_{\Theta_{\mathcal{S}'}}(x)$) if the TDA method works well. The *linear datamodeling score* (LDS) is defined to measure the predictive power of $g_\tau(x, \mathcal{S}'; \mathcal{S})$ and can be formalized as follows:

**Definition E.1** (Linear datamodeling score)**.** Given a training set $\mathcal{S}$, a model output function $f_\Theta$, and a corresponding TDA method $\tau$. Let $\{\mathcal{S}_1, \ldots, \mathcal{S}_m : \mathcal{S}_j \subseteq \mathcal{S}\}$ be $m$ randomly sampled subsets of $\mathcal{S}$, each of size $\alpha \times n$ for some fixed $\alpha \in (0, 1)$. The *linear datamodeling score* (LDS) of $\tau$ for a specific example $x$ is defined as

$$LDS(\tau, x) \triangleq \boldsymbol{\rho}(\{f_{\Theta_{\mathcal{S}_j}}(x) : j \in [m]\}, \{g_\tau(x, \mathcal{S}_j; \mathcal{S}) : j \in [m]\}),$$

---

[2]If a TDA method is additive, then it defines an attribution score that the overall influence of a group is the sum of the individual influence in the group.

where $\boldsymbol{\rho}$ is the Spearman rank correlation (Spearman, 1904), $f_{\Theta_{\mathcal{S}_j}}$ is the model output function with $\Theta_{\mathcal{S}_j}$ learned from $\mathcal{S}_j$ and $g_\tau(x, \mathcal{S}_j; \mathcal{S})$ is defined in Eq (4).

To compute LDS for our experiment settings, we use 50 models that are independently trained on random subsets with size half of the full dataset (i.e., we set $m = 50$ and $\alpha = 0.5$ in Definition E.1).

## F  Wall-clock time measurements

In this paper, we use the wall-clock time on a single A40 GPU to measure the computational costs if the number of independently trained models (i.e., the value of $I$) can not precisely demonstrate the costs, e.g., the training time cost and serving time cost of LoRA Ensemble.

Here, we define several components that dominate the wall-clock time of different ensemble methods.

- $T_{\text{Train}}$: The time to train a model from scratch.
- $T_{\text{Train, Base}}$: The time to train a base model from scratch for LoRA tuning. Normally speaking, $T_{\text{Train, Base}} < T_{\text{Train}}$.
- $T_{\text{Train, LoRA}}$: The time to fine-tune for one LoRA adapter.
- $T_{\text{Serving}}$: The time to calculate the TDA scores for one trained model *after model training.*
- $T_{\text{Serving, Forward-only}}$: The time to calculate the TDA scores for one trained models *after model training* with shared gradients. (will only be used by Dropout Ensemble(forward only). Normally speaking, $T_{\text{Serving, Forward-only}} < T_{\text{Serving, Forward-only}}$.
- $T_{\text{Serving, LoRA}}$: The time to calculate the TDA scores *after model training* for one LoRA adapter. Normally speaking, $T_{\text{Serving, LoRA}} < T_{\text{Serving}}$.

The total computational cost is approximated and summarized in t:

- Training time cost (naive independent ensemble/Dropout Ensemble/Dropout Ensemble (forward only)): $I \times T_{\text{Train}}$.
- Training time cost (LoRA Ensemble): $I \times T_{\text{Train, Base}} + I \times L \times T_{\text{Train, LoRA}}$.
- Serving time cost (naive independent ensemble): $I \times T_{\text{Serving}}$.
- Serving time cost (Dropout Ensemble): $I \times D \times T_{\text{Serving}}$.
- Serving time cost (Dropout Ensemble (forward only)): $I \times T_{\text{Serving}} + I \times (D-1) \times T_{\text{Serving, Forward-only}}$.
- Serving time cost (LoRA Ensemble): $I \times L \times T_{\text{Serving, LoRA}}$.

## G  Additional experiment for Dropout Ensemble

### G.1  Ablation experiment to random projection

Here, we examine the root of the Dropout Ensemble's performance through an ablation experiment. There are no random factors other than dropout for IF, Grad-Dot, or Grad-Cos, while there is another random factor, i.e., random projection, in TRAK. We perform $D$ dropout-masked passes with dropout enabled, i.e., "Dropout Ensemble" and $D$ dropout-masked passes with dropout disabled, i.e., "Only Random Projection", in Table 1 and calculate the LDS. Random projection does contribute to the accuracy improvement, but the improvement will saturate when $D$ is large.

### G.2  Intermediate checkpoints

Here, we show that Dropout Ensemble ensemble can also be applied to intermediate checkpoints and improve the accuracy in Table 2. Park et al. (2023) states that intermediate checkpoints with fewer epochs can be used for ensembling to reduce the training time cost. We save multiple checkpoints from different epochs of the same training process ($I = 1$).

| | $I$ / $D$ | 1 | 3 | 5 |
|---|---|---|---|---|
| Naive Independent Ensemble | N/A | 0.122 | 0.217 | 0.275 |
| DROPOUT ENSEMBLE Only Random Projection | 10 | **0.249** 0.210 | **0.399** 0.335 | **0.457** 0.398 |
| DROPOUT ENSEMBLE Only Random Projection | 25 | **0.316** 0.217 | **0.458** 0.351 | **0.502** 0.413 |

Table 1: Ablating the contribution of test-time dropout. The experiment is carried out on MNIST+MLP. Note that $D$ is the number of dropout-masked passes, and $I$ is the number of independently trained models.

| Ensembling methods | #ckpts / $D$ | 1 | 3 | 5 | 10 |
|---|---|---|---|---|---|
| Naive Independent Ensemble | N/A | 0.122 | 0.170 | 0.188 | 0.201 |
| DROPOUT ENSEMBLE | 10 | 0.249 | 0.318 | 0.342 | 0.362 |
| DROPOUT ENSEMBLE | 25 | 0.316 | 0.359 | 0.373 | 0.389 |

Table 2: The TDA efficacy of DROPOUT ENSEMBLE on intermediate checkpoints from different epochs of the same training process ($I = 1$). Note that $D$ is the number of dropout-masked passes, and #ckpts is the number of intermediate checkpoints used.

## H   Memory costs of Dropout Ensemble and LoRA Ensemble

Here we record the peak memory usage at serving time for DROPOUT ENSEMBLE (Table 3) and LORA ENSEMBLE (Table 4) applied on TRAK algorithm. The memory of vanilla DROPOUT ENSEMBLE is the same as naive independent ensemble. The meomory of DROPOUT ENSEMBLE (Forward Only) is larger than vanilla DROPOUT ENSEMBLE because some of the cached terms. The memory usage of LORA ENSEMBLE is slightly lower than naive independent ensemble because of the reduction in parameter size.

| Datasets and Models | method variants / $D$ | 3 | 10 | 25 |
|---|---|---|---|---|
| MNIST+MLP | DROPOUT ENSEMBLE | 342M | | |
| | DROPOUT ENSEMBLE (Forward Only) | 636M | 1956M | 4787M |
| CIFAR2+MLP | DROPOUT ENSEMBLE | 344M | | |
| | DROPOUT ENSEMBLE (Forward Only) | 638M | 1966M | 4797M |
| CIFAR2+ResNet9 | DROPOUT ENSEMBLE | 477M | | |
| | DROPOUT ENSEMBLE (Forward Only) | 785M | 2087M | 4877M |
| MAESTRO+MusicTransformer | DROPOUT ENSEMBLE | 538M | | |
| | DROPOUT ENSEMBLE (Forward Only) | 634M | 1529M | 3421M |

Table 3: The peak memory usage of DROPOUT ENSEMBLE and its alternative on TRAK with different numbers of dropout-masked passes ($D$) and the number of independently trained models fixed to 5 ($I = 5$).

## I   Space costs of Dropout Ensemble and LoRA Ensemble

Here we record the parameter count to present the space cost for DROPOUT ENSEMBLE (Table 5) and LORA ENSEMBLE (Table 6).

| Ensembling Methods | $\diagdown \frac{I}{L}$ | 1 | 3 | 5 | 10 |
|---|---|---|---|---|---|
| Naive Independent Ensemble | N/A | 431M | 538M | 538M | 538M |
| LoRA Ensemble | 3 | 404M | 404M | 404M | 404M |
| LoRA Ensemble | 10 | 404M | 404M | 404M | 404M |
| LoRA Ensemble | 25 | 404M | 404M | 404M | 404M |

Table 4: The peak memory usage of LoRA Ensemble and naive independent ensemble applied on TRAK for MusicTransformer trained on MAESTRO dataset. Note that $I$ represents the number of independently trained models, and $L$ is the number of LoRA adapters augmented on each model.

| settings $\diagdown$ 
 $I$(w/ any $D$) | MNIST 
 +MLP | CIFAR-2 
 +MLP | CIFAR-2 
 +ResNet-9 | MAESTRO 
 +MusicTransformer |
|---|---|---|---|---|
| 1 | 0.11M | 0.38M | 4.83M | 13.11M |
| 3 | 0.33M | 1.14M | 14.48M | 39.35M |
| 5 | 0.55M | 1.89M | 24.13M | 65.58M |
| 10 | 1.09M | 3.79M | 48.25M | 131.16M |
| 25 | 2.73M | 9.48M | 120.63M | 327.89M |

Table 5: The space cost (total parameter count) of different experiment settings under different numbers of independently trained models ($I$). Note that Dropout Ensemble will **not** incur any additional storage cost with more dropout-masked passes (i.e., the space cost is fixed with respect to $D$).

| $\diagdown \frac{L}{I}$ | Naive Independent 
 ensemble ($L = 0$) | 3 | 10 | 25 |
|---|---|---|---|---|
| 1 | 13.11M | 13.41M | 14.09M | 15.57M |
| 3 | 39.35M | 40.23M | 42.29M | 46.72M |
| 5 | 65.58M | 67.05M | 70.49M | 77.87M |
| 10 | 131.76M | 134.11M | 140.99M | 155.73M |

Table 6: The space cost (total parameter count) of MusicTransformer under different numbers of independently trained models ($I$) and different numbers of LoRA adapters ($L$). LoRA Ensemble only add marginal space cost (at most a 18.7% increment for $D = 25$ across all $I$) compared to naive independent ensembling.

## J Proofs

### J.1 Proof of Lemma 5.1

*Proof.* We proceed by analyzing the bias and variance contributions for both the naive ensemble and the two-step ensemble. We first analyze the bias of the two estimators. Let $\mu_i = \mathbb{E}[\tau(\Theta^{(i)})]$ and $\mu_{(k,d)} = \mathbb{E}[\tau(\Theta^{(k,d)})]$ represent the expected values of the individual and variant estimators, respectively. Since we assume that each individual estimate $\tau(\Theta^{(i)})$ and $\tau(\Theta^{(k,d)})$ have the same and identical distribution, we have $\mu_i = \mu_{(k,d)}$, and thus

$$\text{Bias}^2(\tau_{\text{ens}}) = \left( \frac{1}{I} \sum_{i=1}^{I} \mu_i - \tau(\Theta^*) \right)^2 = \left( \frac{1}{KD} \sum_{k=1}^{K} \sum_{d=1}^{D} \mu_{(k,d)} - \tau(\Theta^*) \right)^2 = \text{Bias}^2(\tau_{\text{2-step}}) \tag{5}$$

Therefore, the difference in squared error $\Delta$ arises solely from the variance terms.

We then analyze the variance of the two estimators. For the naive ensemble estimator $\tau_{\text{ens}}$, the variance is given by:

$$\text{Var}(\tau_{\text{ens}}) = \frac{1}{I^2} \sum_{i=1}^{I} \text{Var}(\tau(\Theta^{(i)})) + \frac{2}{I^2} \sum_{i<j} \text{Cov}(\tau(\Theta^{(i)}), \tau(\Theta^{(j)})) \tag{6}$$

Using the simplified notation, where $\Sigma_{1,1} = \text{Var}(\tau(\Theta^{(i)}))$ and $\Sigma_{i,j} = \text{Cov}(\tau(\Theta^{(i)}), \tau(\Theta^{(j)}))$ for $i \neq j$, this can be written as:

$$\text{Var}(\tau_{\text{ens}}) = \frac{1}{I} \left( \Sigma_{1,1} + (I-1)\Sigma_{i,j} \right) \tag{7}$$

For the two-step ensemble estimator $\tau_{\text{2-step}}$, the variance is given by:

$$\text{Var}(\tau_{\text{2-step}}) = \frac{1}{(KD)^2} \sum_{k=1}^{K} \sum_{d=1}^{D} \text{Var}(\tau(\Theta^{(k,d)})) + \frac{2}{(KD)^2} \sum_{(k,d)<(k',d')} \text{Cov}(\tau(\Theta^{(k,d)}), \tau(\Theta^{(k',d')})) \tag{8}$$

This variance can be decomposed into within-group and between-group covariances. Let $\Sigma_{(k,m),(k,n)} = \text{Cov}(\tau(\Theta^{(k,m)}), \tau(\Theta^{(k,n)}))$ represent the within-group covariance (i.e., between variants of the same base estimator), and let $\Sigma_{(k,m),(l,n)} = \text{Cov}(\tau(\Theta^{(k,m)}), \tau(\Theta^{(l,n)}))$ represents the between-group covariance (i.e., between variants of different base estimators). The variance simplifies to:

$$\text{Var}(\tau_{\text{2-step}}) = \frac{1}{KD} \left( \Sigma_{(k,m),(k,m)} + (D-1)\Sigma_{(k,m),(k,n)} + D(K-1)\Sigma_{(k,m),(l,n)} \right) \tag{9}$$

Finally, the difference in variance between the two estimators, denoted as $\Delta$, is:

$$\Delta = \text{Var}(\tau_{\text{ens}}) - \text{Var}(\tau_{\text{2-step}}) \tag{10}$$

Substituting the variance expressions for $\tau_{\text{ens}}$ and $\tau_{\text{2-step}}$, we have:

$$\Delta = \frac{1}{I} \left( \Sigma_{1,1} + (I-1)\Sigma_{i,j} \right) - \frac{1}{KD} \left( \Sigma_{(k,m),(k,m)} + (D-1)\Sigma_{(k,m),(k,n)} + D(K-1)\Sigma_{(k,m),(l,n)} \right) \tag{11}$$

This expression shows that the difference in error depends on the number of estimators ($I$ in the naive ensemble and $KD$ in the two-step ensemble) and the covariance structure among the estimators. The two-step ensemble will outperform the naive ensemble if the within-group and between-group covariance are sufficiently small compared to the overall covariance in the naive ensemble. $\qquad\square$

## J.2 Derivation of $KD = I$

**Case 2:** $KD = I$ means the total number of estimators in the two-step ensemble $K \cdot D$ is equal to the number of estimators $I$ in the regular ensemble. Thus, both ensembles have the same number of estimates, but the two-step ensemble introduces a base-variant relationship. In this case, we derive:

$$\Delta = \frac{D-1}{I} \left( \Sigma_{i,j} - \Sigma_{(k,m),(k,n)} \right). \tag{12}$$

This case is about comparing the within-group covariance $\Sigma_{(k,m),(k,n)}$ and the individual estimator covariance $\Sigma_{i,j}$. In general, $\Sigma_{(k,m),(k,n)}$ is expected to be larger than $\Sigma_{i,j}$, because $\Sigma_{(k,m),(k,n)}$ is the covariance between variants of the same base estimator, while $\Sigma_{i,j}$ is the covariance between different estimators. This can lead to negative $\Delta$, indicating that the two-step ensemble is outperformed by the regular ensemble in this case. However, since $KD = I$, $\frac{D-1}{I}$ is close to $\frac{1}{K}$, meaning $\tau_{\text{2-step}}$ is not significantly outperformed by $\tau_{\text{ens}}$ when a large $K$ is set. In summary, when $KD = I$, $\tau_{\text{2-step}}$ can still outperform $\tau_{\text{ens}}$ if the within-group covariance is small. However, if this covariance is large, then $\tau_{\text{2-step}}$'s advantage diminishes, and $\tau_{\text{ens}}$ may perform similarly or better due to the independent nature of its estimators. Even for the latter, two-step ensemble still enjoy drastic efficiency gain as generating more variants for each base estimator is much cheaper, e.g., via dropout or LoRA fine-tuning, than generating more individual estimators from re-training.

### J.3 Derivation of the Error Difference for $K = I$ and $KD = I$

For both derivations, we use the following assumptions:

$$\Sigma_{1,1} = \Sigma_{(k,m),(k,m)} \quad \text{and} \quad \Sigma_{i,j} = \Sigma_{(k,m),(l,n)} \text{ for } i \neq j, k \neq l \tag{13}$$

This first equality implies that the variance of each estimator in both ensemble approaches is the same. The second equality implies that the covariance between different estimators in the naive ensemble is the same as the covariance between different estimator variants in the two-step ensemble. These are reasonable assumptions following the i.d. assumption on the individual estimators.

**Case 1: $K = I$.** Substitute $K = I$ into the expression for $\Delta$ and use the assumption $\Sigma_{1,1} = \Sigma_{(k,m),(k,m)}$ and $\Sigma_{i,j} = \Sigma_{(k,m),(l,n)}$, this simplifies $\Delta$ to:

$$\Delta = \frac{1}{I}\left(\Sigma_{1,1} + (I-1)\Sigma_{i,j}\right) - \frac{1}{ID}\left(\Sigma_{1,1} + (D-1)\Sigma_{(k,m),(k,n)} + D(I-1)\Sigma_{i,j}\right) \tag{14}$$

$$= \frac{1}{I}\Sigma_{1,1} + \frac{I-1}{I}\Sigma_{i,j} - \frac{1}{ID}\Sigma_{1,1} - \frac{D-1}{ID}\Sigma_{(k,m),(k,n)} - \frac{I-1}{I}\Sigma_{i,j} \tag{15}$$

$$= \frac{1}{I}\Sigma_{1,1} - \frac{1}{ID}\Sigma_{1,1} - \frac{D-1}{ID}\Sigma_{(k,m),(k,n)} \tag{16}$$

$$= \frac{D-1}{ID}\Sigma_{1,1} - \frac{D-1}{ID}\Sigma_{(k,m),(k,n)} \tag{17}$$

$$= \frac{D-1}{ID}\left(\Sigma_{1,1} - \Sigma_{(k,m),(k,n)}\right) \tag{18}$$

**Case 2: $KD = I$.** Substitute $KD = I$ into the expression for $\Delta$ and use the assumption $\Sigma_{1,1} = \Sigma_{(k,m),(k,m)}$ and $\Sigma_{i,j} = \Sigma_{(k,m),(l,n)}$, this simplifies $\Delta$ to:

$$\Delta = \frac{1}{I}\left(\Sigma_{1,1} + (I-1)\Sigma_{i,j}\right) - \frac{1}{I}\left(\Sigma_{1,1} + (D-1)\Sigma_{(k,m),(k,n)} + (I-D)\Sigma_{i,j}\right) \tag{19}$$

$$= \frac{1}{I}\Sigma_{1,1} + \frac{I-1}{I}\Sigma_{i,j} - \frac{1}{I}\Sigma_{1,1} - \frac{D-1}{I}\Sigma_{(k,m),(k,n)} - \frac{I-D}{I}\Sigma_{i,j} \tag{20}$$

$$= \frac{I-1}{I}\Sigma_{i,j} - \frac{D-1}{I}\Sigma_{(k,m),(k,n)} - \frac{I-D}{I}\Sigma_{i,j} \tag{21}$$

$$= \frac{D-1}{I}\Sigma_{i,j} - \frac{D-1}{I}\Sigma_{(k,m),(k,n)} \tag{22}$$

$$= \frac{D-1}{I}\left(\Sigma_{i,j} - \Sigma_{(k,m),(k,n)}\right) \tag{23}$$

## K   Additional Experiments

Here we report the results of additional experiments, including a new experiment setting, GPT on the Shakespeare dataset for language modeling task, and two existing settings with a larger scale, MNIST with 60K training images and MAESTRO with 15K training music sequences. As can be seen in Figure 7 and Figure 8 (respectively for DROPOUT ENSEMBLE and LoRA ENSEMBLE), both methods still show clear improvements in these new experiments.

## L   Computation Overhead

As we stated in Section 3.3 and Section 3.4, computation overhead is caused by DROPOUT ENSEMBLE and LoRA ENSEMBLE compared to naive independent ensemble. In this section, we list and record the training and serving time of each ensemble methods to show a comprehensive comparison.

In Table 7, we list the computational overhead by different ensemble methods measured on MusicTransformer with TRAK as an example. The overhead of naive ensemble is proportional to the number of models used

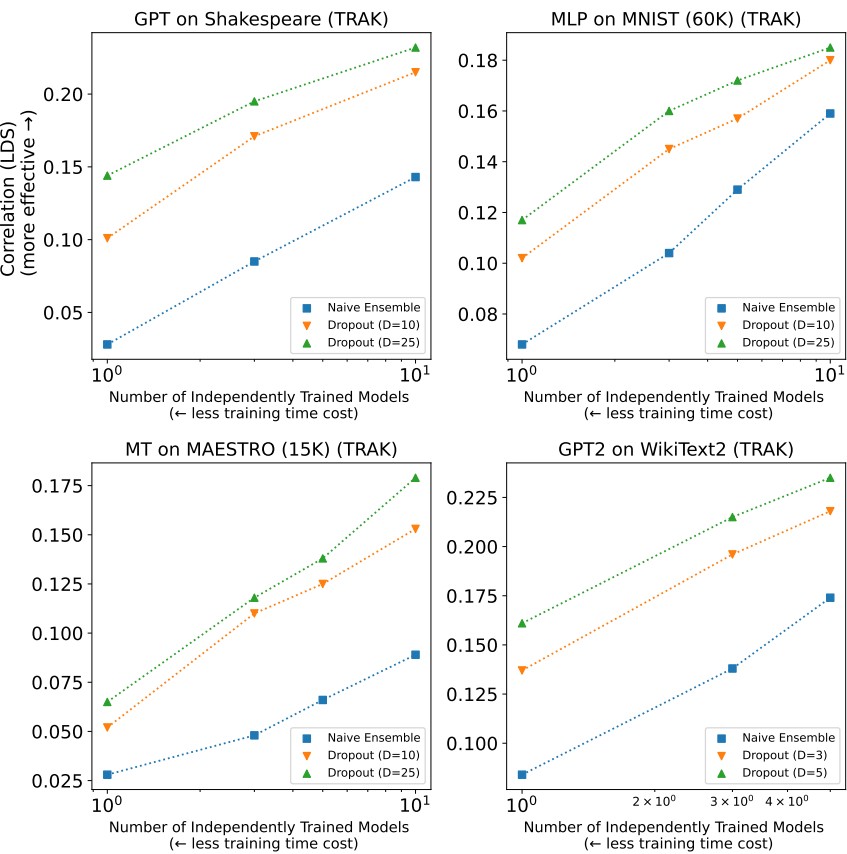

Figure 7: LDS of Naive Ensemble and DROPOUT ENSEMBLE with the same plot format as Figure 3.

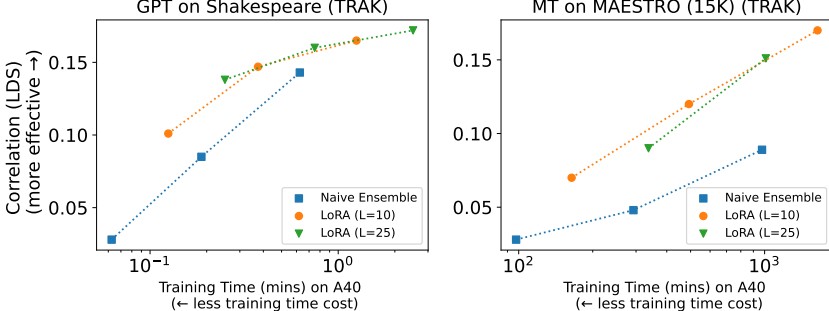

Figure 8: LDS of Naive Ensemble and LoRA ENSEMBLE. Please refer to Figure 6 for the plot format. LoRA ENSEMBLE is applied to Transformer models only therefore the ResNet + MNIST setting is omitted. One point in the MT on MAESTRO (15K) plot is missing as we did not finish the experiment on time.

in the ensemble. In comparison, the DROPOUT ENSEMBLE does not incur any additional cost in terms of training time cost (no additional training) and space cost (the dropout-masked variants do not need to be stored), while incurring additional serving time cost. The LoRA ENSEMBLE incurs additional costs in terms of all three types of costs, but they are much smaller than naive ensemble. Furthermore, the serving time cost is significantly reduced in comparison to DROPOUT ENSEMBLE since we only need to calculate the gradients with respect to the LoRA parameters.

Here we report the computational cost of each efficient ensemble method. The results of the ratio of training over serving cost (i.e. $\frac{\text{training time cost}}{\text{serving time cost}}$) shown in Table 8 demonstrate that training cost dominates the overall computational cost of TDA method.

| Cost Type | No Ensemble | Naive (3) | Naive (10) | D=3 | D=10 | L=3 | L=10 |
|:---:|:---:|:---:|:---:|:---:|:---:|:---:|:---:|
| Training Time | 1 | 3 | 10 | 1 | 1 | 2.00 | 5.50 |
| Serving Time | 1 | 3 | 10 | 3 | 10 | 1.85 | 6.07 |
| Space | 1 | 3 | 10 | 1 | 1 | 1.03 | 1.10 |

Table 7: The relative computation overhead for Naive Ensemble (varying number of ensembles), DROPOUT ENSEMBLE (one base model and varying D), and LoRA ENSEMBLE (one base model and varying L). The "No Ensemble" column refers to the cost with only one model, where the entries are normalized to one for easier comparison. The relative costs of Naive Ensemble and DROPOUT ENSEMBLE are based on simple counting while the relative costs of LoRA ENSEMBLE are based on experiments on MusicTransformer with TRAK.

| Experiment Setting | Training/Serving | Training/Serving(forward-only) |
|:---:|:---:|:---:|
| MNIST-10 + MLP | 40× | 200× |
| CIFAR-2 + MLP | 60× | 450× |
| CIFAR-2 + ResNet9 | 25× | 208× |
| MAESTRO + MusicTransformer | 4.2× | 42× |

Table 8: The training cost / serving cost ratio of each experiment settings.

We also record the training and serving time of each ensemble methods and plot them in Figure 9. The result shows that DROPOUT ENSEMBLE (and the forward-only variant) and LoRA ENSEMBLE can achieve better efficacy-efficiency trade-off. In other words, both ensembles could improve the efficacy with the same computational cost or reduce the computational cost to reach the same efficacy.

Here we also record the serving time of each ensemble methods and plot them in Figure 10. The results shows that DROPOUT ENSEMBLE(forward-only) and LoRA ENSEMBLE could still outperform the naive ensemble if only serving cost is considered. DROPOUT ENSEMBLE has a larger serving time cost overhead. It is worth noting that the training cost and the serving cost are both for the TDA process, which are defined respectively in Section 3.2. The computational cost of TDA should consider both costs together.

## M   Qualitative Results

Here we provide a qualitative results to show how DROPOUT ENSEMBLE improve the TDA result without training more models. We select two random test images from MNIST-10 and the corresponding training samples that are most helpful (with highest score) and most detracting (with lowest score). The result calculated by DROPOUT ENSEMBLE (D=50) is more reasonable by manual inspection. We can observe that the training images identified by DROPOUT ENSEMBLE (D=50) have "shapes" similar to the corresponding test images while helpful (detracting) examples are of the same (different) class as the test images themselves.

## N   Noisy Label Detection Results

The improvement of efficient ensembles could also be demonstrated in downstream tasks of TDA, such as noisy label detection. Here we present the AUC of noisy label detection on MNIST-10 with 20% randomly flipped training samples. The performance of TRAK with DROPOUT ENSEMBLE outperforms the naive independent ensemble without training more models.

## O   Discussion of Dropout Ensemble (forward-only)

DROPOUT ENSEMBLE(forward-only) performs well in the experiments demonstrated in Figure 5 and Figure 9, especially considering the competitive efficacy-efficiency trade-off. A potential explanation to this phenomenon is that the term (Q) affects the performance "substantially" as shown in TRAK (Park et al., 2023)'s ablation

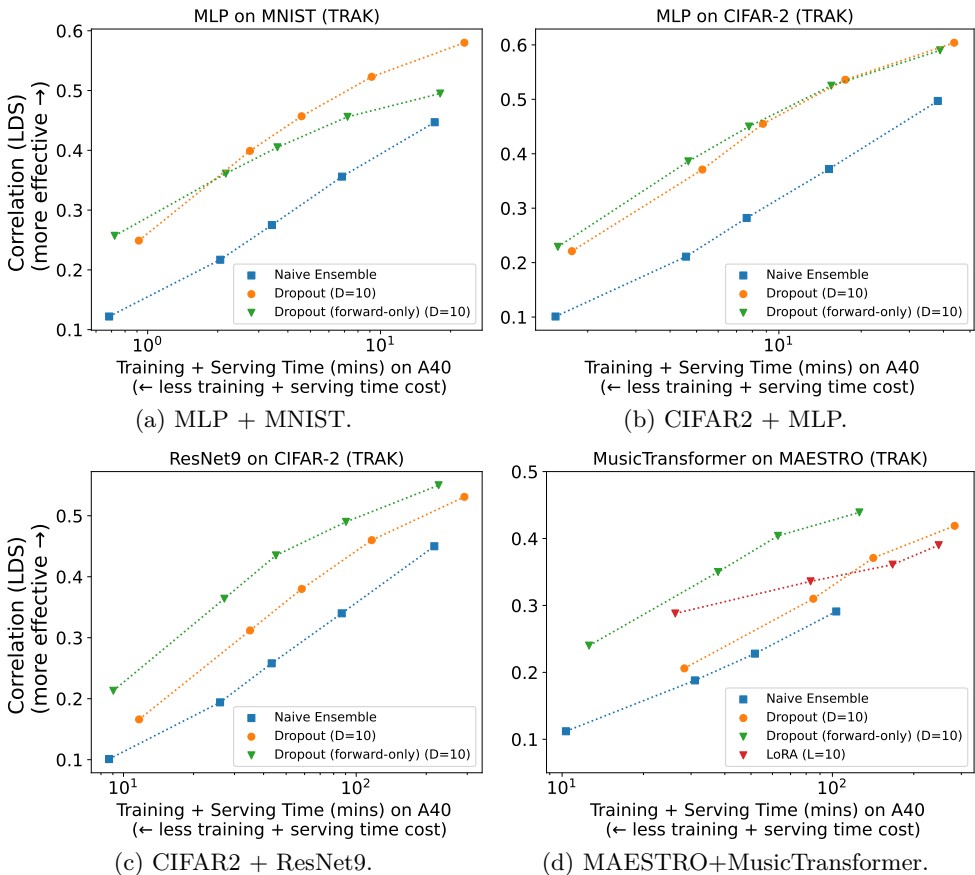

(a) MLP + MNIST.  (b) CIFAR2 + MLP.

(c) CIFAR2 + ResNet9.  (d) MAESTRO+MusicTransformer.

Figure 9: The LDS of naive ensemble, DROPOUT ENSEMBLE, DROPOUT ENSEMBLE (forward-only), and LoRA ENSEMBLE. The x-axis indicates the time costs (training+serving) on a single A40.

| Ensembling methods | #ckpts / $D$ | 1 | 5 |
|---|---|---|---|
| Naive Independent Ensemble | N/A | 0.778 | 0.810 |
| DROPOUT ENSEMBLE | 10 | 0.802 | 0.849 |
|  | 25 | 0.810 | 0.849 |

Table 9: The AUC (Area-Under-Curve) of noisy label detection on MNIST-10 with 20% randomly flipped training samples. TRAK with efficient ensembles can outperform naive TRAK under all settings with drastically fewer computational cost.

study. Focusing on mitigating the effect of randomness on this term and cache other terms could improve the TDA performance effectively and efficiently. It is worth noting that this phenomenon may be closely related to some open questions existing in the TRAK algorithm. For instance, empirical evidence indicates that a particular approach to ensembling models, known as "term-wise" ensemble in the TRAK paper, plays a critical role in TRAK's superior performance. However, the underlying reasons for this remain insufficiently understood.

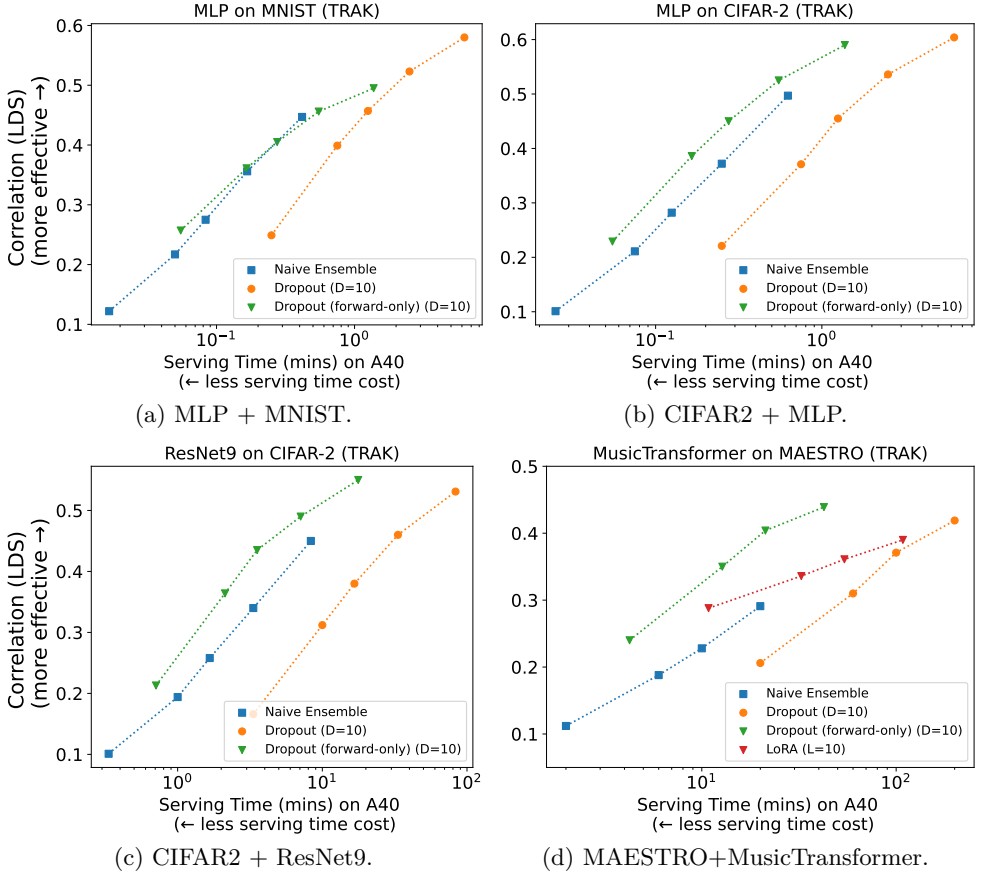

(a) MLP + MNIST.

(b) CIFAR2 + MLP.

(c) CIFAR2 + ResNet9.

(d) MAESTRO+MusicTransformer.

Figure 10: The LDS of naive ensemble, DROPOUT ENSEMBLE, DROPOUT ENSEMBLE (forward-only), and LORA ENSEMBLE. The x-axis indicates the serving time costs on a single A40.

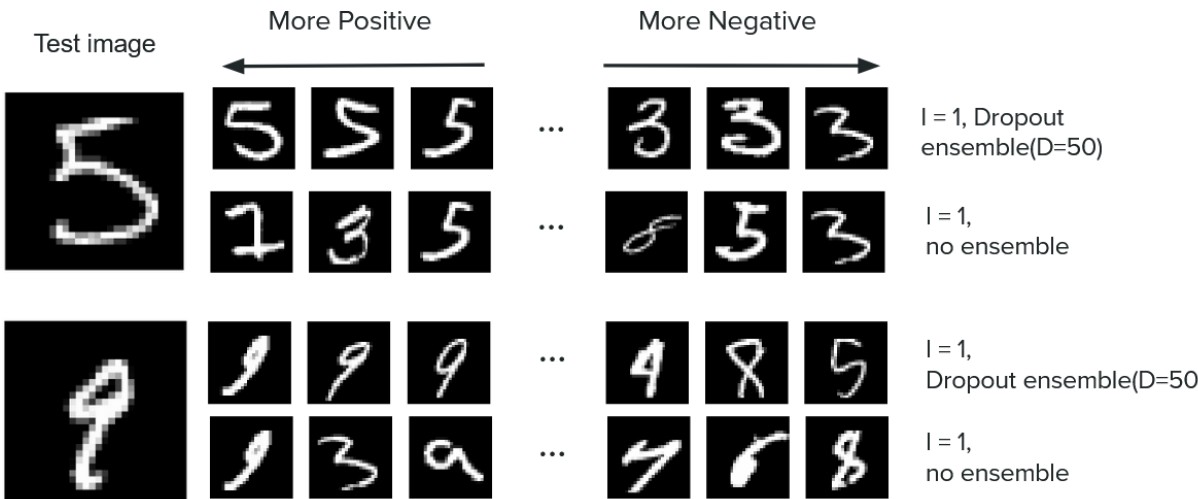

Figure 11: We present two randomly selected test images from MNIST-10 and the training samples that are most helpful (highest score) and most detracting (lowest score) calculated by TRAK with and without the DROPOUT ENSEMBLE.

