# OpenReview forum: "Efficient Ensembling Improves Training Data Attribution"
_TMLR — Rejected by TMLR_

### Review · Reviewer_xdNy · 2025-06-28

**Summary Of Contributions:**

The paper addresses the problem of computational inefficiency in training data attribution (TDA). It proposes two new efficient ensembling strategies: Dropout Ensemble and LoRA Ensemble. These approaches avoid the need to fully retrain many independent models by using dropout masks or lightweight fine-tuning (LoRA), thus reducing training, serving, and space costs significantly. The paper provides theoretical analysis justifying their approach and demonstrates strong empirical results on a range of datasets and models (MNIST, CIFAR-2, MAESTRO).

**Audience:**

Yes

**Claims And Evidence:**

Yes

**Requested Changes:**

* Provide a clearer, systematic comparison between Dropout Ensemble and LoRA Ensemble. Specifically, explain under what conditions one method is preferred over the other (e.g., model size, generative vs. discriminative tasks, hardware constraints), and include experiments or ablation studies to support this guidance.

* Discuss in detail what "sufficiently small" within-group and between-group covariance means in practical terms, and under what situations this assumption might be violated. For example, explain scenarios where models might have high within-group covariance (e.g., when using similar seeds or very similar fine-tuning data) that could undermine the theoretical gains. Provide empirical evidence or diagnostic guidance to help practitioners assess whether the theoretical guarantees apply to their settings.

* Clarify and explicitly detail how the claimed efficiency improvements (e.g., 80% reduction) are computed and realized in practice, since it appears multiple independent models are still trained. Provide concrete breakdowns (e.g., wall-clock times, resource consumption) and compare them clearly to naive ensembling baselines.

**Strengths And Weaknesses:**

Strengths

*  Simplicity and practicality: The proposed methods are simple to implement yet effective, leveraging known techniques (dropout and LoRA) in a novel way for TDA.

* Significant efficiency gains: The experiments show large reductions in training and serving costs (up to 80%) without sacrificing attribution performance (measured by LDS).

* Arch and Dataset Evaluation: Experiments span multiple architectures, datasets.

Weaknesses

*  Partial empirical analysis: While LDS is a useful metric, no additional robustness metrics (e.g., stability to data perturbations, test-time data shift scenarios) are considered. This limits understanding of real-world resilience.

*  Limited analysis of theoretical assumptions: The theoretical analysis states that "the two-step ensemble will outperform the naive ensemble if the within-group and between-group covariance are sufficiently small compared to the overall covariance in the naive ensemble." However, it is unclear what "sufficiently small" means in practice, and under what real-world conditions this assumption may be violated (e.g., in highly non-convex or highly correlated settings). The paper does not analyze when this assumption holds or provide empirical diagnostics to check it, which limits the practical value of the theory.

* Unclear comparative advantages between proposed methods: The paper proposes two methods (Dropout Ensemble and LoRA Ensemble), but it is not clearly explained under what conditions one method is preferred over the other. A systematic comparison or practical guidance on selecting between them would be very valuable.

* Insufficient clarity on actual efficiency gains: While the paper claims up to 80% reduction in training cost, it remains unclear how this is precisely achieved, especially since multiple independent models are still trained. The explanation of cost computation (e.g., wall-clock time, GPU hours) and the specific breakdown versus naive ensembling is not sufficiently detailed.

---

> ### Author Response · Authors · 2025-07-13
>
> **Thank you for acknowledging our work to be practical and brings significant gains in multiple experiment settings**. We reply to the comments as following:
>
> > [W1] Partial empirical analysis
>
> Thank you for the feedback. For the data perturbation case, we have a study in **Appendix N** for noisy label detection, where we perturb 20% of the data labels and leverage TDA methods to do mislabel detection. It shows that our method could lead in mislabel detection ratio performance, a downstream task in a data perturbation scenario.
>
> > [W2 & RC2] Limited analysis of theoretical assumptions
>
> Thank you for the feedback! We want to clarify the relationship of the theoretical analysis and the assumption you mentioned as follows.
>
> The efficacy gap between naive ensemble and two-step is defined and derived in Lemma 5.1. To analyze the gap in a clearer way, we especially focus on two cases for the two-step ensemble of TDA methods.
>
> **The first one is $K=I$**, which means that for both Naive ensemble and two-step ensemble (e.g., Dropout Ensemble and LoRA Ensemble) we use the same number of independent trained models. The gap between Naive ensemble and two-step ensemble is defined by Equation 3. The gap is positive (two-step ensemble is better) as long as $\Sigma_{1,1}$ is larger than $\Sigma_{(k,m)(k,n)}$, **which is likely to be the case for both Dropout and LoRA Ensemble with no strict assumptions.**
>
> **The second one is $KD=I$**, which means that the total number of estimators for Naive ensemble and two-step ensemble are the same (but two-step ensemble methods will be much more efficient since D variants of the second step ensemble can be acquired through Dropout or LoRA). The gap is defined by Equation 12. **We did not claim that the within-group covariance is never significantly larger than the covariance between individual predictors. We actually acknowledged that our advantage will diminish if this covariance gap is huge. However, our goal in Case 2 was to illustrate that our method can help reduce the error significantly, by a factor of 1/K.** In practice, this error reduction helps to maintain a competitive performance while providing a drastic efficiency gain, as reflected in our experiments with very small K <= 5.Specifically for the MLP model on MNIST, a small K = 5 with Dropout Ensemble can outperform the naive ensemble with 25 models.
>
> Including empirical evidence could be a good idea to further clarify the theoretical analysis, we will include them in the final draft for our experiment settings.
>
> > [W3 & RC1] Unclear comparative advantages between proposed methods
>
> Thank you for the suggestion. The main difference that could guide practical guidance among Dropout Ensemble and LoRA Ensemble is their different overhead characteristics, which we have some discussion in **Appendix L.**
>
> > (Appendix L) In comparison, the Dropout Ensemble does not incur any additional cost in terms of training time cost (no additional training) and space cost (the dropout-masked variants do not need to be stored), while incurring additional serving time cost. The LoRA Ensemble ... Furthermore, the serving time cost is significantly reduced in comparison to Dropout Ensemble since we only need to calculate the gradients with respect to the LoRA parameters.
>
> Models with relatively larger parameter size typically have smaller training/serving ratio for TDA cost (see Table 8). **Thus, LoRA ensembles are more preferable for large and transformer-based models. Furthermore, LoRA Ensemble is normally used for Transformer architectures, which normally have larger parameter size and less LoRA implementation effort.** We will rearrange some results and analysis to the main text in our final draft since this could be critical information.
> > [W4 & RC3] Insufficient clarity on actual efficiency gains
>
> Thank you for the suggestion.
>
> - To analyze the cost and overhead for each ensemble method, we split and define the cost as training time cost, serving time cost, and space cost in Section 3.2. We compare the efficiency gain **when achieving the same efficacy**, which can also be translated to efficacy gain with the same efficiency. For example, the 80% training cost improvement can be found on Figure 3 where the efficacy metric (LDS) of Dropout Ensemble with 1 independent training can outperform the Naive Ensemble with 5~10 independent training.
> - We also explain about the wall-clock time measures in **Appendix F** to break down the costs in different categories for each ensemble method. Furthermore, the computation overhead (Table 7) of both efficient and naive ensemble, relative ratio between training & serving cost (Table 8), and a wall-time record of the whole computational cost (Figure 9) is shown and discussed in **Appendix L** to further enhance the understanding.
>
> We will rearrange some results and analysis in our final draft to help readers understand the efficiency gains more clearly.

---

### Review · Reviewer_CHJm · 2025-06-29

**Summary Of Contributions:**

The paper introduces two lightweight ensembling approaches for gradient-based train-data attribution (TDA):

1. Dropout Ensemble: Reuses a single model’s dropout masks at inference to approximate an ensemble, cutting training and storage costs by up to 80%.

2. LoRA Ensemble: Uses low-rank LoRA adapters to spawn multiple “heads” from one base model, achieving similar attribution fidelity to full ensembles with minimal extra parameters.

A theoretical “two-step estimator” framework underpins both methods, and extensive experiments on supervised (MNIST, CIFAR-2) and generative (Music Transformer on MAESTRO) tasks show they match naive ensembling’s effectiveness while drastically reducing compute and memory.

**Audience:**

Yes

**Broader Impact Concerns:**

Since attribution methods inherit and can even amplify existing model biases, deploying these techniques without comprehensive bias audits and privacy safeguards may exacerbate unfair or unethical outcomes.

**Claims And Evidence:**

Yes

**Requested Changes:**

1. Evaluate Dropout and LoRA ensembles on at least one high-capacity model to demonstrate scalability and stability in realistic settings.

2. I'd like to see the authors' discussion on the following points since it's not clear to me yet:
- Since gradient-based methods generally struggle with complex non-convex models, I'm wondering whether the proposed ensemble approaches can mitigate or compound this brittleness. Have you observed failure modes or stability differences compared to naive ensembling?
- For Section 3.3 - TRAK’s Optimization, can you provide intuition or empirical analysis explaining why the optimized approach still preserves attribution fidelity?

**Strengths And Weaknesses:**

Strengths:
- The dropout and LoRA strategies are practical, low-overhead ensemble designs that require no separate model training and can be immediately applied to existing pipelines. The empirical results show that they are simple and effective.
- The empirical evaluation spans diverse model families and attribution metrics, and the results are strong.

Weaknesses:
- Core evaluations use relatively small toy subsets and models. It remains unclear how well these ensembling methods hold up on large vision or language models, which is the primary scenario this work aims to enable.

---

> ### Author Response · Authors · 2025-07-13
>
> **Thank you for acknowledging our work to be practical and effective on multiple experiment settings.** We reply to the comments as following:
>
> > [W1, RC1] Scalability to large dataset and large models.
>
> In **Appendix K**, we have several larger experiment settings. Two new experiment settings on GPT architecture (up to **124M parameters**) and Shakespeare/WikiText dataset are used. Furthermore, it includes full datasets for MNIST and MAESTRO (60K and 15K training data points). The results show that both Dropout Ensemble and LoRA Ensemble could be scaled to high-capacity models as well as large data size with significantly better performance than Naive Ensemble.
>
> > [RC 2.1] non-convex brittleness
>
> TDA ensemble aims to **mitigate the randomness of non-convex model training**, which is a common issue for nearly all gradient-based TDA methods like influence function and TRAK, as shown in Figure 4, the performance of multiple TDA methods could be improved through ensemble methods. But other shortcomings lie in the gradient-based method formula such as singularity of hessian matrices, non-convergence training and Taylor approximation error is at least not directly mitigated by ensemble methods.
>
> > [RC 2.2] TRAK’s Optimization
>
> We have some discussion about this in **Appendix O**. In short, we think a potential explanation is the significant effect of term (Q) to the TDA performance in TRAK’s ablation study. However, this is still an open question in the TRAK algorithm and stated in TRAK paper, which remains to be insufficiently understood.
>
> > Broader Impact Concerns
>
> Thank you for the insightful feedback about broader impact. Though TDA is only an analysis to the model and does not change the model, the interpretation derived by TDA may involve some kind of bias. We will have some discussion about this in our final draft.

---

### Review · Reviewer_vhkT · 2025-07-04

**Summary Of Contributions:**

This paper introduces two efficient ensemble strategies using Dropout Ensemble and LoRA Ensemble to improve gradient-based Training Data Attribution (TDA). These methods replace costly independent ensembling with lightweight approximations that maintain attribution quality while significantly reducing training (up to 80%), serving (up to 60%), and storage costs (up to 80%). The authors validate their methods across diverse TDA techniques, model types, and tasks (including generative modeling), and provide theoretical analysis showing reduced estimation error.

**Audience:**

Yes

**Broader Impact Concerns:**

TDA can surface sensitive training data or raise copyright concerns in generative models. The paper should explicitly discuss privacy and fairness risks, especially in the context of model transparency and data ownership.

**Claims And Evidence:**

Yes

**Requested Changes:**

1. Test LoRA Ensemble on a classification task.
2. Summarize key theoretical results more intuitively.
3. How does ensemble sizes influence the results? should be discussed

**Strengths And Weaknesses:**

Strengths:
1. Significant efficiency gains with minimal loss in TDA performance.
2. Works across multiple datasets, models, and TDA methods.
3. Backed by theoretical justification.
4. Applicable to both classification and generative settings.

Weaknesses:
1. LoRA is only tested on generative tasks.
2. Since there are multiple ensembling
3. Apart from ensemble, there are different efficient ways to mimic the variance, such as [1]. I wonder could works such as [1] also works in TDA task. In other words, I think it is better to include more comparison methods other than dropout ensemble
4. In such analysis, the author combine dropout and lora as efficient ways to get variants, does this mean that we can have more different combinations. Paper should discuss the probability.

[1] Consistency Calibration: Improving Uncertainty Calibration via Consistency among Perturbed Neighbors

---

> ### Author Response · Authors · 2025-07-13
>
> Thank you for acknowledging our work to be practical and effective in multiple experiment settings, with theoretical justification. We reply to the comments as following:
>
> > [W1, RC1] LoRA Ensemble on classification tasks
>
> Thank you for the advice. **For many cases, LoRA is used on transformer architecture for generative tasks, so we focused on generative tasks (GPT and MusicTransformer) to show its practical usage.** We have added an experiment setting of classification settings for LoRA Ensemble, LoRA Ensemble can still improve the TDA performance with much less overhead compared to Naive Ensemble for classification task.
> |                      | I=1    | I=3    |
> |----------------------|--------|--------|
> | Naive                | 0.0278 | 0.0464 |
> | LoRA Ensemble (L=5)  | 0.0415 | 0.0499 |
> | LoRA Ensemble (L=10) | 0.0432 | 0.0541 |
>
> *Exp setting*: For this additional experiment, we construct a CIFAR-10 split by uniformly sampling 500 images per class for training (5000 total) and 50 per class for testing (500 total). We adopt a compact Vision Transformer (ViT-Base/16) variant with 4 transformer layers, hidden size 256, 4 attention heads, feed-forward layer dimensionality 768, which results in 2.88 million parameters in total. We train this model from scratch with the AdamW optimizer (learning rate 1e-3, weight decay 2e-4), a batch size of 512, and run for 70 epochs, applying a simple data pipeline of 224×224 resize and normalization.
> We apply the LoRA ensemble protocol to the aforementioned ViTmodel with the CIFAR-10 dataset. First, we train $I$ independent ViT backbones from scratch for 30 epochs on the full 5000-image training set using the same AdamW schedule (lr = 1e-3, weight decay = 2e-4) and data pipeline described above. Next, for each of these base models, we inject LoRA adapters of rank r=8 and LoRA scaling α=8 (with trainable biases only) into every query/value matrix within all transformer layers, leaving the rest of the ViT parameters frozen. For each base model, we then draw $L$ random half-subsets of the 5000-image training data (size = 2500 each) and fine-tune each LoRA-augmented model for an additional 30 epochs (AdamW, lr =5e-3, weight decay = 1e-5). This yields a total of $I\times L$ lightweight LoRA-fine-tuned models (each with only ~0.04 M additional trainable parameters).
>
> > [W2]
>
> It seems that this review comment is not complete, we are happy to reply to any further questions once the reviewer completes this review comment.
>
> > [W3, W4] Potential alternative efficient ensemble strategies
>
> Thank you for the insightful suggestion! We agree there are potentially some other alternative efficient ensemble strategies besides Dropout Ensemble and LoRA Ensemble. Some methods like the one in [1] could be developed and extended to other efficient ensemble strategies. We will discuss more in the paper and inspire further studies in this direction.
>
> [1] Consistency Calibration: Improving Uncertainty Calibration via Consistency among Perturbed Neighbors
>
> > [RC2] Intuitive summary to the theory
>
> Our key theoretical results are
> 1) We derive the difference in error to the “perfect score” of the TDA estimator is related to the covariance of each base estimator.
> 2) Through the derived error formula, we found that 2-step (first independent train $K$ estimators; second efficiently get $D$ estimators) ensemble, compared to independently trained $I$ estimators like naive ensemble, have the advantage in providing a better trade-off between performance and efficiency.
> 3) We specifically analyze two cases. For $K=I$, a 2-step ensemble will have the advantage of a naive ensemble with small computational overhead. For $KD=I$, a 2-step ensemble could significantly reduce the gap with a naive ensemble with small K and improve the performance-efficiency trade-off.
> Thank you for the suggestion, we will include a more clear intuitive explanation in our theory section.
>
> > [RC 3] Ensemble size affect the result
>
> Larger ensemble size helps the performance and the improvement gradually saturates. The trend is true for both naive ensemble and efficient ensemble methods, while efficient ensemble could significantly help the efficiency.  This can be found in Figure 3 and 6.
>
> > Broader impact concerns
>
> Thank you for the feedback on broader concerns. TDA methods are model analysis methods and require training data samples as input. It is sometimes used to help understand or resolve the copyright and other model transparency/data ownership issues. Without the training data being exposed to the attributor, TDA (in the format we defined in Section 3.1) could be difficult to perform. We will include some discussion in our final draft.

---

### Decision · Action_Editor_sM2x · 2025-08-12

**Recommendation:** Reject

**Additional Comments:**

All reviewers leaned toward acceptance. However, I find some of Reviewer xdNy’s comments to be important for the main claims (please see the Claims and Evidence section), and this reviewer also wished that the comments could be addressed despite leaning toward acceptance. Furthermore, I think that addressing all of these important comments amounts to more than a minor revision. For this reason, I have to recommend rejection at this time with encouragement to resubmit, so that the changes to the manuscript can be fully reviewed.

The changes that I think would be important are as follows:
- Add to the text describing Figure 3 and similar figures to make clear where the headline efficiency gains are achieved (i.e., which curves, which points). Annotating the figures may help.
- Present all of the cost metrics (training, serving, storage) in the main paper to give the full picture. This may involve moving material from Appendices F and L. For Dropout Ensemble, this means the additional serving cost in addition to the training and storage cost savings shown in the main text. For LoRA Ensemble, more justification should be given for the reductions reported in Section 4.3.
- Discuss when the theoretical assumptions regarding different covariances are expected to hold and when they might be violated. Empirical computation of the covariances would be a plus.

The authors should also take the opportunity to address other reviewer comments. In particular:
- For Reviewer CHJm, the author rebuttal points to additional results in Appendix K for larger GPT models. However, I did not find a description of the setup for this experiment like for the other experiments.
    - As I was skimming the appendix to check on the last point, I happened to notice that Figure 11 and Table 9 (and possibly others) are not referenced in the text. Please check that the appendix reads well.
- For Reviewer vhkT, the rebuttal reports on an additional experiment using LoRA Ensemble for image classification, which would be good to include.

**Audience:**

Yes

**Audience Explanation:**

This submission improves the trade-off between efficacy and efficiency for gradient-based TDA methods that use an ensemble of models. Reviewers agree that this is an important contribution.

**Claims And Evidence:**

No

**Claims Explanation:**

This submission is on the use of ensembles of models to improve the efficacy of gradient-based training data attribution (TDA) methods. The main contribution of the paper is to show that two efficient ensembling methods, Dropout Ensemble and LoRA Ensemble, have similar efficacy as the naive ensembling of fully independent models while reducing the costs of training, serving, and storing the models. The paper also provides theoretical analysis to help understand when the proposed ensembling methods should outperform naive ensembling.

All reviewers find that the main empirical claim is well-supported in terms of the tasks (image classification, generative music and language modelling), corresponding models, and gradient-based TDA methods that are evaluated. The reviewers also find the efficiency gains to be significant at a high level. The simplicity and practicality of the proposed methods are also praised.

Reviewer xdNy comments that the main paper is insufficiently clear about how (and where) the headline efficiency gains are achieved, and about the breakdown in terms of training, serving, and storage costs. The authors’ response points to existing results in Figure 3, Appendix F, and Appendix L, and promises to rearrange these results to clarify the efficiency gains. However, the current version of the manuscript does not show these revisions.

Reviewer xdNy also notes that the paper does not adequately discuss when the theoretical assumptions are expected to hold (more precisely, the degree to which they are expected to hold). In my view, understanding the theoretical assumptions is important because of the claim that the theory helps with understanding. In their rebuttal, the authors reiterate the assumption made for the case $K = I$ presented in the main paper (between-estimator variance is greater than within-group covariance), but do not elaborate on why this assumption should hold “with no strict assumptions.” For both cases $K = I$ and $KD = I$ (the latter in the appendix), the authors mention that empirical evidence would bolster the theoretical assumptions. I agree that empirical computation of the involved covariances would help and it seems to be feasible, but it remains to be done.

**Resubmission Of Major Revision:**

The authors may consider submitting a major revision at a later time.